# Image-based screen capturing misfolding status of Niemann-Pick type C1 identifies potential candidates for chaperone drugs

Ryuta Shioi[1], Fumika Karaki[1¤a], Hiromasa Yoshioka[1], Tomomi Noguchi-Yachide[1], Minoru Ishikawa[2], Kosuke Dodo[3], Yuichi Hashimoto[1], Mikiko Sodeoka[3], Kenji Ohgane[1,3¤b]*

1 Institute for Quantitative Biosciences, The University of Tokyo, Bunkyo-ku, Tokyo, Japan, 2 Graduate School of Life Sciences, Tohoku University, Aoba-ku, Sendai, Japan, 3 Synthetic Organic Chemistry Laboratory, RIKEN Cluster for Pioneering Research, Wako, Saitama, Japan

¤a Current address: Laboratory of Medicinal Chemistry, School of Pharmacy, Kitasato University, Shirokane, Minato-ku, Tokyo, Japan
¤b Current address: Department of Applied Bioscience, Faculty of Science and Technology, Tokyo University of Science, Noda, Chiba, Japan
* ohgane@rs.tus.ac.jp

**Data Availability Statement:** The raw image files obtained from the screening are available from the Dryad repository (https://doi.org/10.5061/dryad.fn2z34tpt). Other relevant data, including R scripts

## Abstract

Niemann-Pick disease type C is a rare, fatal neurodegenerative disorder characterized by massive intracellular accumulation of cholesterol. In most cases, loss-of-function mutations in the *NPC1* gene that encodes lysosomal cholesterol transporter NPC1 are responsible for the disease, and more than half of the mutations are considered to interfere with the biogenesis or folding of the protein. We previously identified a series of oxysterol derivatives and phenanthridine-6-one derivatives as pharmacological chaperones, i.e., small molecules that can rescue folding-defective phenotypes of mutated NPC1, opening up an avenue to develop chaperone therapy for Niemann-Pick disease type C. Here, we present an improved image-based screen for NPC1 chaperones and we describe its application for drug-repurposing screening. We identified some azole antifungals, including itraconazole and posaconazole, and a kinase inhibitor, lapatinib, as probable pharmacological chaperones. A photo-crosslinking study confirmed direct binding of itraconazole to a representative folding-defective mutant protein, NPC1-I1061T. Competitive photo-crosslinking experiments suggested that oxysterol-based chaperones and itraconazole share the same or adjacent binding site(s), and the sensitivity of the crosslinking to P691S mutation in the sterol-sensing domain supports the hypothesis that their binding sites are located near this domain. Although the azoles were less effective in reducing cholesterol accumulation than the oxysterol-derived chaperones or an HDAC inhibitor, LBH-589, our findings should offer new starting points for medicinal chemistry efforts to develop better pharmacological chaperones for NPC1.

for analysis, raw uncropped gel images, and quantitative data used for preparing the figures are available from Mendeley Data (https://doi.org/10.17632/jr23ccpp46).

**Funding:** The work described in this article was supported in part by Incentive Research Grant (2014) for K.O. from RIKEN (https://www.riken.jp/en/), JSPS Grant-in-Aid for Young Scientists B for K.O. (JSPS KAKENHI Grant number 17K15487) and JSPS Grants-in-Aid for Scientific Research B for Y.H. (JSPS KAKENHI Grant number JP17H03996) from Japan Society for the Promotion of Science (https://www.jsps.go.jp/english/index.html). The funders had no role in study design, data collection and analysis, decision to publish, or preparation of the manuscript.

## Introduction

Niemann-Pick disease type C is a fatal, neurodegenerative disorder characterized by massive accumulation of cholesterol in late endosomes/lysosomes (LE/L). About 95% of cases harbor mutations in the Niemann-Pick type C1 (*NPC1*) gene [1, 2], and the others harbor mutations in Niemann-Pick type C2 (*NPC2*) gene [3]. These genes encode for LE/L proteins that are essential for intracellular cholesterol transport. Specifically, cholesterol delivered to cells as low-density lipoprotein particles (LDL) is captured by soluble NPC2 protein in the LE/L, and NPC2 hands off the bound cholesterol to the N-terminal sterol binding domain of NPC1 protein, a large 13-pass transmembrane protein in the LE/L [4–7]. Then NPC1 transfers cholesterol across the LE/L membrane to intracellular compartments via currently unknown mechanisms [7, 8].

Loss of function of either NPC1 or NPC2 results in accumulation of cholesterol and other glycosphingolipids in the LE/L, which contributes to the pathology of NPC. More than 100 mutations of NPC1 have been described, and most are considered to interfere with its proper folding and stability [9–12]. For example, a well-characterized mutation, I1061T (Ile to Thr mutation at residue 1061, **Fig 1A**), slows down the folding process of the protein at the endoplasmic reticulum (ER) without affecting its cholesterol-transporting function [13]. The folding intermediate of NPC1 mutant is retained in the ER by the cellular quality-control machinery and eventually degraded via ER-associated degradation (ERAD), leading to loss of functional NPC1 protein in the LE/L compartment [14].

We previously identified a class of oxysterol derivatives as pharmacological chaperones (PCs) for NPC1 (**Fig 1B**). We showed that these compounds stabilize and enhance folding of the folding-defective mutant via direct binding to the protein [15–19], and alleviate cholesterol accumulation in patient-derived fibroblasts having NPC1 with the I1061T mutation [20, 21]. Although our subsequent search for more drug-like NPC1 chaperones yielded a class of nonsteroidal PCs for NPC1, we could only obtain compounds with more than one-order-of-magnitude lower activity, even after medicinal chemistry work [22].

On the other hand, an approach called drug-repurposing is gaining traction, in which already-approved drugs of known safety and pharmacokinetic profile are screened and developed for treatment of new diseases, especially for rare diseases such as Niemann-Pick disease type C [23]. As safety and pharmacokinetic issues are major bottlenecks in the drug development process, searching for approved drugs that also act as PCs for NPC1 is a very promising strategy.

In this study, we aimed to identify potential NPC1 chaperones from a library of approved drugs. For this purpose, we developed an image-based chaperone assay for NPC1, and performed a drug-repurposing screen. We successfully identified several classes of compounds that improve the proteostasis of NPC1 mutant, including several potential PCs, as well as previously reported proteostasis modulators.

## Materials and methods

### Library, reagents, buffers, and plasmids

A library of FDA-approved drugs (Screen-Well FDA-approved drug library V2 version 1.0), containing 768 drugs was purchased from Enzo Life Sciences. Chemical reagents used in this study and their commercial sources were as follows: VCP inhibitor CB5083 (Cayman), n-dodecyl-beta-D-maltoside (DDM, Dojindo), itraconazole (TCI), posaconazole (TCI), ketoconazole (TCI), ravuconazole (TCI), terconazole (Sigma-Aldrich), miconazole nitrate (TCI), hydroxyitraconazole (Toronto Research Chemicals), imatinib (TCI), lapatinib (Cayman

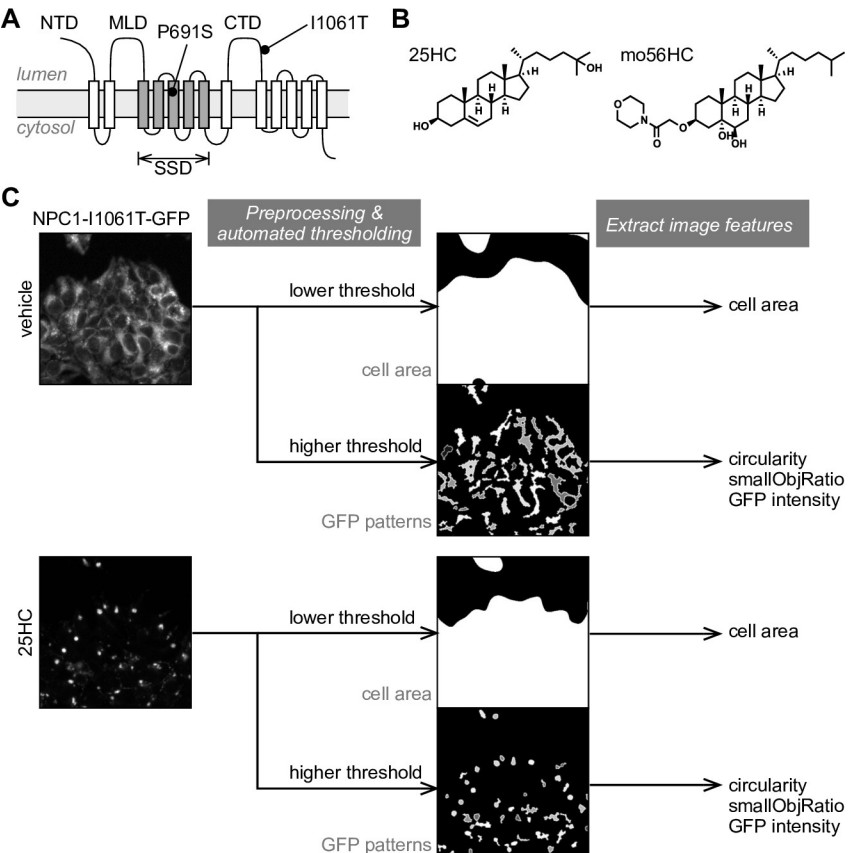

**Fig 1. An image-based screen for chaperone drugs that correct mis-localization of NPC1-I1061T-GFP.** (A) Schematic representation of the topology of NPC1 and the positions of P691S and I1061T mutations in the protein. The following abbreviations are used to denote the domains of NPC1; NTD, N-terminal domain; MLD, mid luminal domain; SSD, sterol-sensing domain; CTD, C-terminal luminal domain. (B) Structures of oxysterol-based pharmacological chaperones for NPC1-I1061T mutant protein. (C) Illustration of the image processing procedure used to extract the NPC1-I1061T-GFP localization pattern and relevant image features. From the resulting images, the cell area and GFP distribution patterns are extracted by applying automated thresholding. From the GFP patterns, morphological features that reflect localization changes from ER to LE/L, including circularity and smallObjRatio, were calculated and used for quantifying the chaperone effect.

Chemical), digoxin (TCI), progesterone (TCI). Filipin complex (F9765) and 25-hydroxycholesterol (25HC) were obtained from Sigma-Aldrich.

The following buffers were used: Dulbecco's phosphate-buffered saline (PBS), PBST (PBS supplemented with 0.05% Tween 20), tris-buffered saline (TBS, 20 mM Tris-HCl pH 7.5 and 137 mM NaCl), TBST (TBS supplemented with 0.05% Tween 20), TNET (25 mM Tris-HCl pH 7.4, 150 mM NaCl, 5 mM EDTA, and 1% Triton X-100), TNED (25 mM Tris-HCl pH 7.4, 150 mM NaCl, 5 mM EDTA, and 0.5% DDM).

The plasmid for expression of FLAG-tagged ΔNTD-NPC1-P691S/I1061T-tGFP was prepared by introducing P691S mutation into pCMV-ΔNTD-NPC1-I1061T-tGFP. Site-directed mutagenesis was performed by PCR mutagenesis with KOD plus high-fidelity polymerase, DpnI, and Ligation high ver.2 (all from Toyobo). The following primers were used for the site-directed mutagenesis: forward primer, `ATCTCGTTCCTGGTGCTGGCTGTTG` (from `CCG` to `TCG`); reverse primer, `GACTTCAATCACAATGAGGGTCAAG`.

Synthetic procedures and characterization data of photo-crosslinking probes (itraAZY, itraACT, itraBIO, and lapaAZY) are available in **S1 Appendix**.

## Cell culture, transfections, and stable cell lines

HEK293 cell lines stably expressing wild-type (WT), I1061T, ΔNTD, ΔNTD-I1061T mutants of FLAG-tagged NPC1-FLAG-tGFP (NPC1-GFPs) under the control of CMV promoter were described previously [20], and were cultured in high-glucose DMEM (FUJIFILM Wako Pure Chemicals) supplemented with 5% heat-inactivated fetal bovine serum (FBS) at 37˚C in a humidified incubator (5% $CO_2$).

To establish HEK293 stably expressing ΔNTD-NPC1-P691S/I1061T-GFP, HEK293 cells at around 80% confluence were transfected with the plasmid using Lipofectamine LTX (Invitrogen) transfection reagents, and selection pressure was applied with 0.4 mg/mL G418 sulfate. Single clones were isolated by limiting dilution and expanded.

## Image-based screen

Cells stably expressing NPC1-I1061T-GFP were cultured on poly-D-Lys-coated glass-bottomed 96-well plates (#164588, Nunc) or CELLSTAR μClear 96-well plates (#655090, Greiner), and treated with vehicle (0.1% DMSO) or test compounds (10 μM) at 50–70% confluence. After 20–21 h, cells were fixed with 10% formalin-PBS containing 1 μg/μL Hoechst 33342 for 30 min, and washed with PBS. Images were acquired on an IN Cell Analyzer 2000 (GE Healthcare) with the following settings; 60x (air) objective lens, 2x2 binning, Quad1 dichroic mirror and DAPI/FITC filters, with hardware autofocus, plate heater 30˚C, and exposure time for DAPI 0.060 sec and FITC channel 0.450 sec. Six fields of images separated by 200–300 μm were acquired for each well. Processing of the acquired images and extraction of image features were performed with *EBImage*, an R package for image processing [24]. From a pair of GFP and Hoechst images, we extracted the NPC1-GFP pattern (from GFP images), cell area (from GFP images), and the number and area of nuclei (from Hoechst images) as illustrated in **Fig 1C**. To extract the NPC1-GFP pattern, we applied the following image processing sequence: gamma transform (gamma = 0.5), subtraction of gaussian-blurred image (sigma = 35) as a background subtraction, adaptive thresholding (kernel size = 25, offset = 0.02), closing operation (disc-shaped kernel with size = 7) to the thresholded binary image, and removal of too small objects (size < 50). From the raw GFP image and the extracted GFP pattern, GFP intensity within the extracted pattern, NPC1-GFP area, and shape-related features, including circularity, were calculated. To extract cell area from the GFP image, we applied the following image processing sequence: gamma transform (gamma = 0.1) to reduce the contribution from high-intensity areas, gaussian blur (sigma = 25), and Otsu's thresholding to obtain a binary image for overall cell area. To calculate the number and area of nuclei, the following image processing sequence was applied: mean filter (disc-shaped kernel with size = 11), gamma transform (gamma = 0.6), Otsu's thresholding, segmentation (distance map transform and watershed segmentation), filter objects by size (1,000 < size < 7,000) to remove too large or too small objects, and calculate the number and area of nuclei. The calculated image features along with the treatment conditions are available from the Mendeley Data repository (http://dx.doi.org/10.17632/jr23ccpp46.3). The raw images obtained from the screen are available from the Dryad repository (https://doi.org/10.5061/dryad.fn2z34tpt), and an example R script for the image processing is also available from the Mendeley Data repository (http://dx.doi.org/10.17632/jr23ccpp46.3).

## Flow-cytometric analysis of expression levels of NPC1-GFPs

Cells grown on 12-well plates were treated as indicated at 70–90% confluence, and incubated for 20–21 h. The cells were then detached with 0.25% trypsin-EDTA, and suspended in culture medium, and the GFP fluorescence from 10,000 cells was recorded on a FACS Canto II (BD

Biosciences). The data was subjected to logicle transformation for visualization [25]. Data analysis and visualization were performed with R (version 3.3.3) along with RStudio (version 1.0.153) using the *flowCore* and *flowViz* packages. The raw data is available from the Mendeley Data repository (http://dx.doi.org/10.17632/jr23ccpp46.3).

## Clear-native PAGE analysis of NPC1-GFPs

Clear-native polyacrylamide gel electrophoresis (CN-PAGE) was run as reported [26] with slight modifications. Cells stably expressing the indicated NPC1-GFPs were treated with vehicle (0.1% DMSO) or test compounds for 20–21 h. The cells were washed with PBS, and lysed with TNED buffer supplemented with cOmplete protease inhibitor cocktail (EASYpack EDTA-free, Roche) on ice for 1 h. Cell debris was removed by centrifugation (14,000 g, 10 min, 4˚C) and the supernatant was collected. The concentration of the lysate was normalized based on total protein concentration as determined by BCA protein assay. The same amount of the lysate (27 μg total protein / lane) was mixed with 0.5 volume of 3x loading buffer (0.01% DOSS, 0.05% Ponceau S, 60% glycerol, and 50 mM Tris-HCl pH 8.6), and loaded on 5–12% SuperSep Ace gel (Wako). CN-PAGE was performed at 0˚C with cathodic buffer (0.02% DDM, 0.005% DOSS, 5% glycerol, 25 mM Tris, and 192 mM glycine) and anodic buffer (25 mM Tris and 192 mM glycine) at constant current (15 mA) for 90–100 min. The gel was briefly washed with water, and GFP fluorescence was imaged on a FLA7000 gel imager (without ND filter, 50 μm grid). Gel images were processed with Image J for visualization (cropping and linear contrast adjustment) and for quantification. For quantification, the lane profile was first extracted from the images after background correction (rolling ball algorithm). The lane profiles were then plotted and peak area was calculated using R in combination with *hyperSpec* and *baseline* packages. To stain total protein in the native-PAGE gels, we applied 2,2,2-trichloroethanol (TCE) staining [27]; the native-PAGE gels after GFP detection were soaked in 10% TCE in 50% aqueous methanol for 10 min at room temperature with gentle shaking. The gels were briefly rinsed with water and irradiated with UV (302 nm) for 2 min on a UV transilluminator (UVP dual intensity UV transilluminator). Fluorescence images were acquired on a EzCapture cooled CCD camera imager equipped with UV transilluminator (302 nm) and 560 nm long-pass filter (YA-3) filter. Note that no recognizable GFP signal could be observed at the top of the gels under our experimental condition, implying that terminally misfolded NPC1 proteins were not soluble in the lysis buffer, if any.

## Ubiquitination assay

Confluent cells on poly-D-Lys-coated 6-well plates were treated with 3 μM CB5083, which inhibits Valosin-containing protein (VCP; required for extraction and subsequent proteasomal degradation of ubiquitinated membrane proteins in the ER [28]) and the indicated compounds for 6 h. Then the cells were washed once with PBS, and lysed with TNED buffer supplemented with cOmplete EDTA-free protease inhibitor cocktail (Roche) and 10 mM N-ethylmaleimide on ice for 1 h. The lysate was centrifuged to remove cell debris. The total protein concentration was determined by BCA protein assay and adjusted to 2.0 μg/μL. The diluted lysate (300 μL) was mixed with pre-washed anti-FLAG M2 magnetic beads (Sigma-Aldrich, M8823, 4 μL bed volume/sample) and the mixture was rotated overnight at 4˚C. The beads were washed twice with 300 μL of TNED buffer, and eluted with 30 μL of 1×Laemmli sample buffer by heating at 60˚C for 15 min. The samples (10 μL/lane) were resolved by SDS-PAGE (40 mA, 45 min, in an ice-water bath) using 7.5% SuperSep Ace gel (Wako). The proteins were transferred to PVDF membrane (Immobilon P, Millipore), blocked with 1% BSA-TBST (r.t., 1 h), and stained overnight with anti-ubiquitin antibody conjugated with HRP

(1/2,000 dilution, P4D1, mouse monoclonal antibody, sc-8017, Santa Cruz Biotechnology) [29] in signal enhancer, CanGetSignal solution 2 (Toyobo). The membrane was washed with TBST and bands were detected with Immobilon Western chemiluminescence reagent (Millipore) using an EzCapture imaging system (ATTO). The blotted membrane was washed with water, and incubated with a stripping buffer (2% SDS, 62.5 mM Tris-HCl pH 6.8, 100 mM 3-mercapto-1,2-propanediol) at 50˚C for 30 min. The stripped blot was washed with water and TBST, and blocked again with 1% BSA-TBST for 1 h. The membrane was washed with TBST, and incubated with anti-FLAG M2 antibody (Sigma-Aldrich, F1804, 1/2,000 dilution) [20] diluted in CanGetSignal solution 1 for 1 h. After five washes in TBST, the membrane was shaken with goat polyclonal anti-mouse IgG conjugated with HRP (Millipore, 12–349) [20] diluted in CanGetSignal 2 (1/2,000 dilution) for 1 h. The membrane was washed five times and chemiluminescence detection was performed as described above. The quantification of the ubiquitination level and NPC1 level was performed as described above for CN-PAGE data analysis.

## Photoaffinity labeling

Photoaffinity labeling of NPC1 was performed as described previously [20] with slight modifications. Confluent cells stably expressing the indicated NPC1-FLAG-GFP construct on a 10 cm dish were washed with PBS (2 mL) and scraped into cold PBS (1 mL). The cells were centrifuged (300×g, 5 min, 4˚C) and the pellet was resuspended in TSE buffer (20 mM Tris-HCl pH 7.6, 250 mM sucrose, and 1 mM EDTA) supplemented with cOmplete EDTA-free protease inhibitor cocktail (Roche). The cells were disrupted with a probe sonicator (USP-300, Shimadzu) at 0˚C, and nuclei and cell debris were removed by centrifugation (2,000×g, 5 min, 4˚C). The supernatant was centrifuged (125,000×g, 1 h, 4˚C) to pellet membranes. The pelleted membranes were resuspended in TBS and total protein concentration was determined by BCA protein assay; the result was used to adjust the concentration to 4.0 μg/μL with TBS. The membrane was aliquoted onto a U-bottomed polypropylene 96-well plate (Greiner, 650261) and treated with the indicated photoaffinity labeling probe (1/100 volume, 50 μM stock solution in DMSO) and a competitor (1/100 volume, 5 mM stock solution in DMSO) or DMSO, and incubated for 30 min on ice. The samples were irradiated with UV light (Handy UV lamp, UVP, 365 nm, ~1 cm from the sample) on ice for 20 min. The irradiated membranes were solubilized by adding 10% (v/v) SDS-10% (v/v) Triton X-100 solution (1/10 volume) and incubated at 0˚C for 30 min. Separately, a 10×catalyst solution for the click reaction (10 mM $CuSO_4$, 1 mM TBTA, 50 mM aminoguanidine, and 50 mM sodium ascorbate) was prepared as follows [30]: 100 mM $CuSO_4$ (12.5 μL), water (11.3 μL), 1.7 mM TBTA in DMSO/$t$BuOH (4:1) (7.5 μL), 200 mM aqueous aminoguanidine (31.3 μL), and freshly prepared 100 mM aqueous sodium ascorbate (62.5 μL) were mixed and vortexed to obtain a colorless heterogenous solution. To the solubilized membrane was added biotin-PEG3-azide (Tokyo Chemical Industry, 1.25 μL of 5 mM stock solution in DMSO, 50 μM), and then the 10×catalyst solution (12.5 μL) was added. The reaction was performed at room temperature for 1 h. Then, the reaction mixture was diluted 10-fold with TNET, and immunoprecipitated with anti-FLAG M2 magnetic beads (4 μL bed volume / sample, prewashed with TBS and TNED) at 4˚C overnight. The magnetic beads were washed twice with TBS (400 μL) and once with TNET (400 μL), and eluted with 1×Laemmli sample buffer (30 μL/sample) at 60˚C for 15 min. The eluates were resolved by SDS-PAGE (SuperSep Ace 7.5% gel, Wako) and bands were transferred to PVDF membrane (Immobilon P PVDF membrane, Millipore). The membrane was blocked with 1% (w/v) BSA-TBST (0.05% Tween 20 in TBS) at 4˚C overnight, and probed for biotin with ImmunoPure streptavidin HRP (1/2,000 in TBST, Pierce, 21126) [20] for 1.5 h at room temperature.

The membrane was washed and chemiluminescence detection was performed as described above for ubiquitination assay. The membrane was stripped and re-probed with anti-FLAG M2 antibody as described above. Quantification of the bands was performed with an ImageJ macro, BandPeakQuantification tool [31].

### LAMP1 immunostaining and colocalization analysis

Stable cell lines expressing NPC1-GFPs were grown to 50–70% confluence on single-well glass-bottomed dishes (IWAKI, No. 3911–035) coated with poly-D-Lys, and treated as indicated for 20–21 h. The cells were washed once with PBS and fixed with 10% formalin-PBS at room temperature for 30 min. The fixed cells were permeabilized with 0.1% Triton X-100 in PBS, blocked with 1% BSA-PBST, and stained with mouse monoclonal anti-LAMP1 antibody (Abcam, H4A3, ab25630) [20] at a dilution of 1:100 in PBST for 2 h. The cells were washed with PBST, and incubated with anti-mouse IgG conjugated with Alexa546 (ThermoFisher, A-11003) [20] diluted with PBST at 1:1000 for 1 h. After three washes with PBST, images of NPC1-GFPs and the Alexa548-stained LAMP1 were acquired on an FV3000 confocal laser microscope (×60 oil-immersion objective lens, UPLSAPO 60XS2). Colocalization analysis was performed in the R environment (version 3.3.3) using the *EBImage* package [24] and visualized with the *ggplot2* package. Differences between the groups were evaluated by applying the Kruskal-Wallis rank sum test and Dunn's multiple comparison with the Benjamini-Hochberg method for adjustment of the false discovery rate, using an R package, *dunn.test*. The raw images used and the corresponding R script are available from the Mendeley Data repository (http://dx.doi.org/10.17632/jr23ccpp46.3).

### Filipin staining

NPC fibroblasts (I1061T/I1061T, GM18453, obtained from the NIGMS Human Genetic Cell Repository at the Coriell Institute for Medical Research) were grown on glass-bottomed 96-well plates (#164588, Nunc) and treated as indicated. Filipin staining for evaluation of cholesterol accumulation was performed as previously described [20], and images were acquired on an IN Cell Analyzer 2000 using the following settings: 60x (air) objective lens, 2x2 binning, Quad1 dichroic mirror and DAPI/DIC filters, with hardware autofocus, plate heater 30˚C, and exposure time for DAPI (filipin) 0.100 sec and bright field 0.100 sec. Nine fields of images separated by 300 μm were acquired for each well. Quantification of relative filipin intensity, defined as filipin intensity within high-intensity lysosome-like organelles per total filipin intensity within cells, was performed using R and the *EBImage* package [24]. Details of the image processing are summarized in **S1 Fig**, and an example R script for the image analysis is available from the Mendeley Data repository (http://dx.doi.org/10.17632/jr23ccpp46.3).

## Results

### Development of an image-based screen to identify compounds that improve the folding efficiency of the NPC1-I1061T mutant

To evaluate the chaperone-like activity of test compounds towards the NPC1-I1061T mutant, we exploited the subcellular localization of NPC1-I1061T-GFP. This folding-defective NPC1 mutant is recognized by the cellular quality control machineries and retained in the ER for eventual degradation via the ERAD pathway [13, 14, 20]. Thus, at the steady state, the NPC1-I1061T-GFP mutant shows an ER localization pattern. In the presence of proteostasis regulators, including PCs, the folding intermediate of the NPC1 mutant is stabilized to a more native-like conformational state, and is allowed to traffic to the LE/L compartment, similarly

to wild-type (WT) NPC1-GFP. Our previous method to evaluate this localization change involved colocalization analysis with immunostained lysosomal-associated membrane protein 1 (LAMP1), a marker of the LE/L compartment [20], or simply visual phenotypic classification [21]. Although both methods offered a reliable measure for evaluation of PC activity, their signal-to-noise ratio and throughput were insufficient for large-scale screening.

To overcome these problems, we developed an automatable, image-based screen that quantifies the localization pattern of NPC1-I1061T-GFP in terms of morphological features. In this screen, time-consuming and costly immunostaining is not required, and only NPC1-I1061T-GFP images and Hoechst (nuclear stain) images are acquired. The distribution pattern was extracted from the GFP images by application of image processing techniques, and the morphology of the pattern was quantified in terms of several selected features, which were confirmed to reflect the ER-to-LE/L change of NPC1-I1061T localization in our preliminary experiments (**Fig 1C**). In addition, the approximate cell area was extracted from the GFP images by applying a lower threshold, and the number of nuclei and average nuclear size were also obtained from the Hoechst images as measures of cytotoxicity (**Fig 1C**).

Among the image features we tested, "circularity" and "smallObjRatio" gave high sensitivity and S/N ratio. The circularity is defined as the average ratio of the area of each object to the area of a circle with the same perimeter ($4^*\pi^*area/perimeter^2$) for each NPC1-GFP-positive object, and this feature gives higher values for circular objects and lower values for longer, branched objects. A smallObjRatio is defined here as the area ratio of NPC1-GFP objects with less than a specified threshold size with respect to the total NPC1-GFP-positive area. The good performance of these parameters is intuitively understandable, considering that the localization of NPC1-I1061T-GFP changes from network-like, larger patterns of ER localization, to vesicular, smaller patterns of LE/L localization. We also confirmed that the expression level of the mutant protein, measured as GFP fluorescence intensity per cell area (GFPInt), exhibited a dose-dependent increase, as previously reported [20]. As these three features capture different aspects of the chaperone-induced localization change, we envisioned that multiparametric selection of hit compounds would increase the sensitivity and accuracy of the hit selection process. Of the three features, the circularity feature generally gave the highest Z' factor of >0.5 for samples treated with 0.3–3 μM mo56HC, a previously developed NPC1 chaperone, indicating suitability of the screen for large-scale screening and for quantitative comparison of potency [32]. Although all three features showed a clear dose-response relationship for mo56HC, GFPInt was much noisier and less sensitive than the other morphological features (**Fig 2B**). Thus, we established an image-processing protocol that yields several good measures of chaperone activity towards the NPC1-I1061T mutant.

## Screening of a library of approved drugs

With the validated image-analysis protocol in hand, we used it to perform drug-repurposing screening of 768 approved drugs in order to identify compounds that affect the folding efficiency or proteostasis status of the NPC1-I1061T mutant. The image features obtained from the screen were first normalized to plate medians in a plate-by-plate manner, and transformed into a robust Z score, which is defined as the effect size (values–plate median) divided by the median absolute deviation (MAD, a non-parametric version of SD) of the plate [33]. The positive control mo56HC, placed on each plate, gave a reproducible dose-dependent effect in terms of circularity, smallObjRatio, and GFPInt (**Fig 2A and 2B**), and again the circularity measure gave the clearest separation from negative controls. Note that cells began to round up at 10 μM mo56HC due to cytotoxicity, resulting in a lower smallObjRatio, while circularity was less affected.

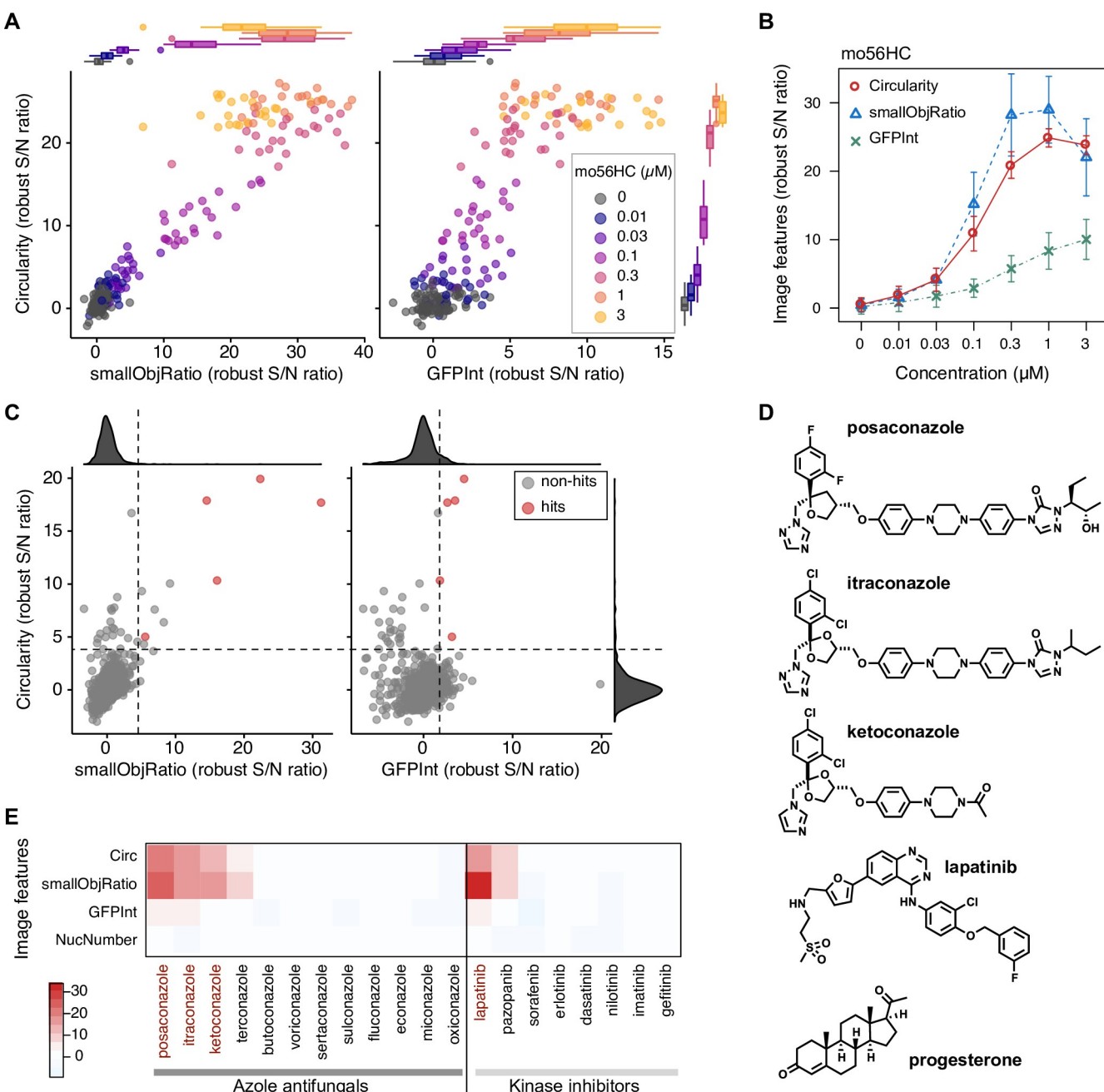

**Fig 2. Screening of an FDA-approved drug library identified several classes of potential NPC1 chaperones.** (A) Three image features (circularity, smallObjRatio, and GFPInt) capture the chaperone-mediated localization change of NPC1-I1061T-GFP. Data points are from a series of mo56HC-treated wells on 11 screening plates (duplicate dose-response analysis per plate). The image features are presented in a set of 2D plots and shown as robust S/N ratio (effect size normalized with median absolute deviation) after plate-wise normalization with respect to plate medians. (B) Dose-dependency of mo56HC examined by employing each of the three image features. Circularity gave the best separation of negative and positive data. Data points and error bars represent mean ± SD (n = 11 experiments, each performed in duplicate). (C) Screening results represented in a set of 2D plots (circularity, smallObjRatio, and GFPInt). Screening was performed at 10 μM. The dashed lines denote thresholds for each of the image features, and compounds that exceeded all the thresholds were selected as hits (red circles). (D) Chemical structures of the hits. (E) Heatmap representation of the screening results for the selected hits and closely related compounds in the library. The heatmap colors encode the robust S/N ratio of image features.

For hit selection, we used multiparameter thresholding of the circularity, smallObjRatio, and GFPInt. The thresholds were determined by taking median values for the features (circularity 4.03; smallObjRatio 4.01; GFPInt 1.50) from a marginally active positive control group (0.03 μM mo56HC). With this threshold, no negative controls (vehicles on each plate and blank wells on the library plates) were identified as hits (0 of 76 negative controls), and 5 hit compounds were identified: itraconazole, posaconazole, ketoconazole, lapatinib, and progesterone (**Fig 2C and 2D** and **S2 Fig**).

Three of the hit compounds (itraconazole, posaconazole, and ketoconazole) are azole antifungals, which preferentially inhibit fungal lanosterol demethylase, a cytochrome P450 enzyme, along with human CYP enzymes. Lapatinib is a kinase inhibitor that primarily targets endothelial growth factor receptors (EGFRs). The other compound was a steroidal hormone, progesterone, which we previously identified during our SAR study of oxysterol-based chaperones, so we excluded this from further analysis [21]. Notably, the chaperone-like activity was specific for a subset of the azole antifungals and kinase inhibitors, indicating that the activity is not mediated through CYPs or kinases (**Fig 2E**). Also, comparison of the azole antifungals in the library suggested the importance of a specific substructure, N,N-diaryl piperazine, for chaperone activity against the NPC1 mutant.

In addition to these hit compounds, we noted that the proteasome inhibitor bortezomib and the $Na^+,K^+$-ATPase inhibitor digoxin exhibited significantly higher circularity values, despite their high toxicity (lower cell area and fewer nuclei) under the screening conditions (10 μM treatment) (**S2** and **S3 Figs**). In contrast to mo56HC or the hit compounds, which gave high smallObjRatio values, bortezomib gave a low value for this parameter. Inspection of the raw images revealed that bortezomib treatment resulted in accumulation of NPC1-I1061T-GFP protein in an aggresome-like structure, consistent with the potent proteasome-inhibitory activity of bortezomib (**S2 Fig**) [34]. Although images from digoxin-treated cells contained only small numbers of cells due to significant toxicity (**S3 Fig**), the localization of NPC1-I1061T-GFP within the remaining cells implied a distribution pattern distinct from that of untreated cells. Therefore, we included digoxin for further validation.

## Azoles and lapatinib behave similarly to oxysterol-based pharmacological chaperones

To validate the obtained hit compounds, we first performed dose-response analysis of the hits and related compounds (**Fig 3A and 3B**). Among the azoles tested, itraconazole (and its primary metabolite hydroxyitraconazole) and posaconazole exhibited potent activity, comparable to that of mo56HC. Ketoconazole and ravuconazole were only weakly active, and the other azoles, including terconazole and miconazole, were inactive. Lapatinib, a kinase inhibitor, was confirmed to be active at 10 μM, but was much less potent than the azoles, while the other kinase inhibitors, including imatinib and erlotinib, were inactive. Interestingly, digoxin exhibited an activity profile distinct from those of itraconazole, lapatinib, or mo56HC; while these three compounds increased the GFP intensity in addition to the circularity feature, digoxin reduced the GFP intensity in a dose-dependent manner, implying that digoxin acts differently from mo56HC, azoles, or lapatinib.

To further validate the hit compounds, we examined their effect on the steady-state expression level of NPC1-I1061T-GFP, expression of which is driven by the CMV promoter. As previously demonstrated, pharmacological chaperones upregulate NPC1-I1061T-GFP, probably via stabilization of the destabilized mutant protein [20]. Flow-cytometric analysis clearly confirmed an up-regulatory effect of the azoles (itraconazole, OH-itraconazole, posaconazole, and ketoconazole) and lapatinib, and the dose-response relationships were consistent with those

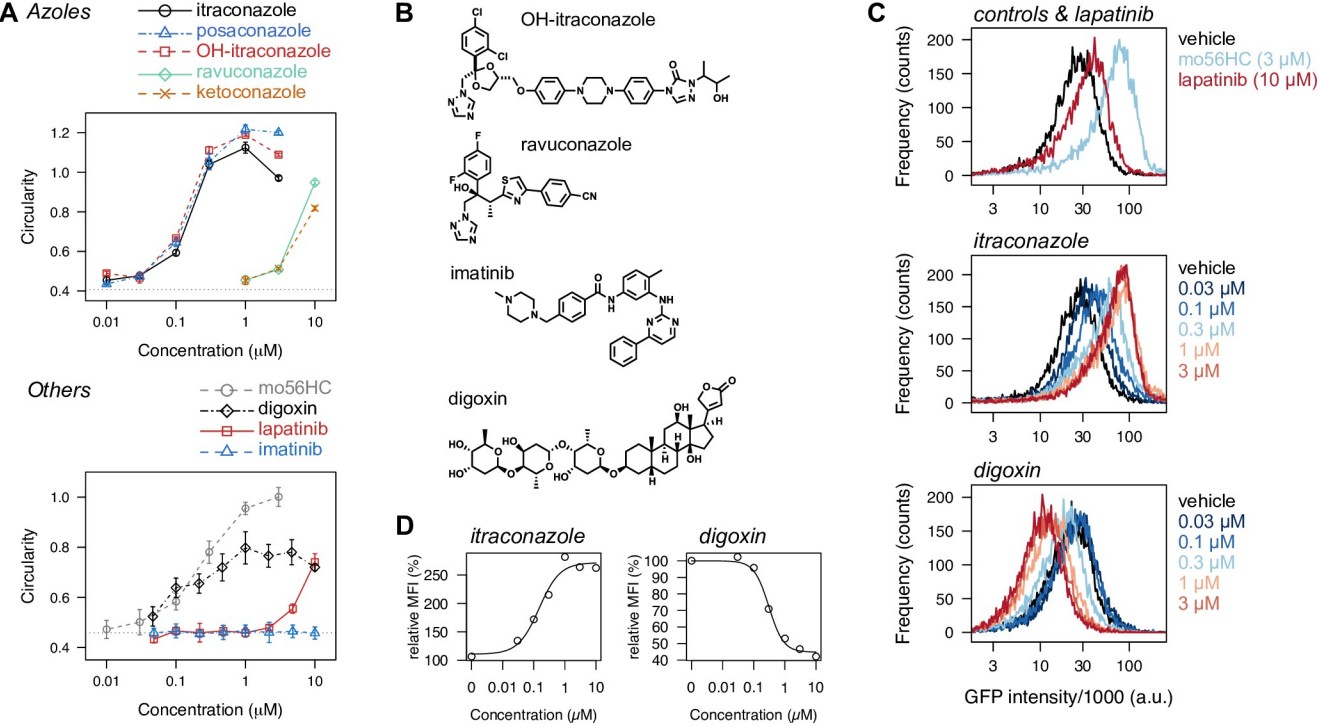

**Fig 3. Validation of the hits by dose-response analysis and flow-cytometric analysis.** (A) Dose-response analysis of the hits and related compounds. (B) Chemical structures of the compounds used in (A) and not shown in Fig 2. (C) Stabilizing effect of the hits on the NPC1 mutant, examined by flow-cytometric analysis. Cells stably expressing NPC1-I1061T-GFP were treated as indicated for 21 h, and processed for flow-cytometric analysis to quantify the steady-state expression level of the mutant protein. Consistent with the stabilizing effect of mo56HC (upper panel), lapatinib (upper panel) and itraconazole (middle panel) both stabilized the mutant protein. (D) Dose-dependent increase in the mean GFP fluorescence intensity from the flow-cytometric data shown in (C). Itraconazole (left panel) increased the NPC1-I1061T-GFP level approximately 270% with an $EC_{50}$ of 0.15 μM, and digoxin (right panel) reduced the expression level to 45% with an $EC_{50}$ of 0.31 μM.

obtained from the image-based assay, supporting the idea that these compounds act as pharmacological chaperones for NPC1 (**Fig 3C and 3D**). Consistent with the idea, the up-regulatory effect was also observed in the presence of emetine, a protein synthesis inhibitor, supporting our hypothesis that the hits attenuate the degradation rate of the NPC1 mutant (**S4 Fig**) [28, 35, 36]. Digoxin was also confirmed to decrease the expression level of NPC1-I1061T-GFP at sub-micromolar concentrations. This different profile of digoxin clearly supports a distinct action mechanism for digoxin from those of the other hit compounds.

## Clear-native PAGE analysis supports improved folding status of NPC1-I1061T mutant in the presence of the hit compounds

In the above experiments, the instability of the NPC1-I1061T mutant was captured in terms of the steady-state expression level of the mutant protein in the cells. We also examined this point by clear-native polyacrylamide gel electrophoresis (CN-PAGE), which allows evaluation of the hydrodynamic status of GFP-fused proteins under non-denaturing conditions by means of in-gel fluorescence detection, similarly to fluorescence-detection size-exclusion chromatography [26, 37, 38]. Cells expressing either WT- or NPC1-I1061T-GFP were treated as indicated, and their lysates were subjected to CN-PAGE analysis, which utilizes a mixture of a mild anionic detergent, dioctyl sulfosuccinate (DOSS), and a nonionic detergent, dodecyl maltoside (DDM). Note that we also tested another CN-PAGE protocol that relies on a colorless derivative of Coomassie Brilliant Blue (mCBB) and obtained comparable results, but we selected the

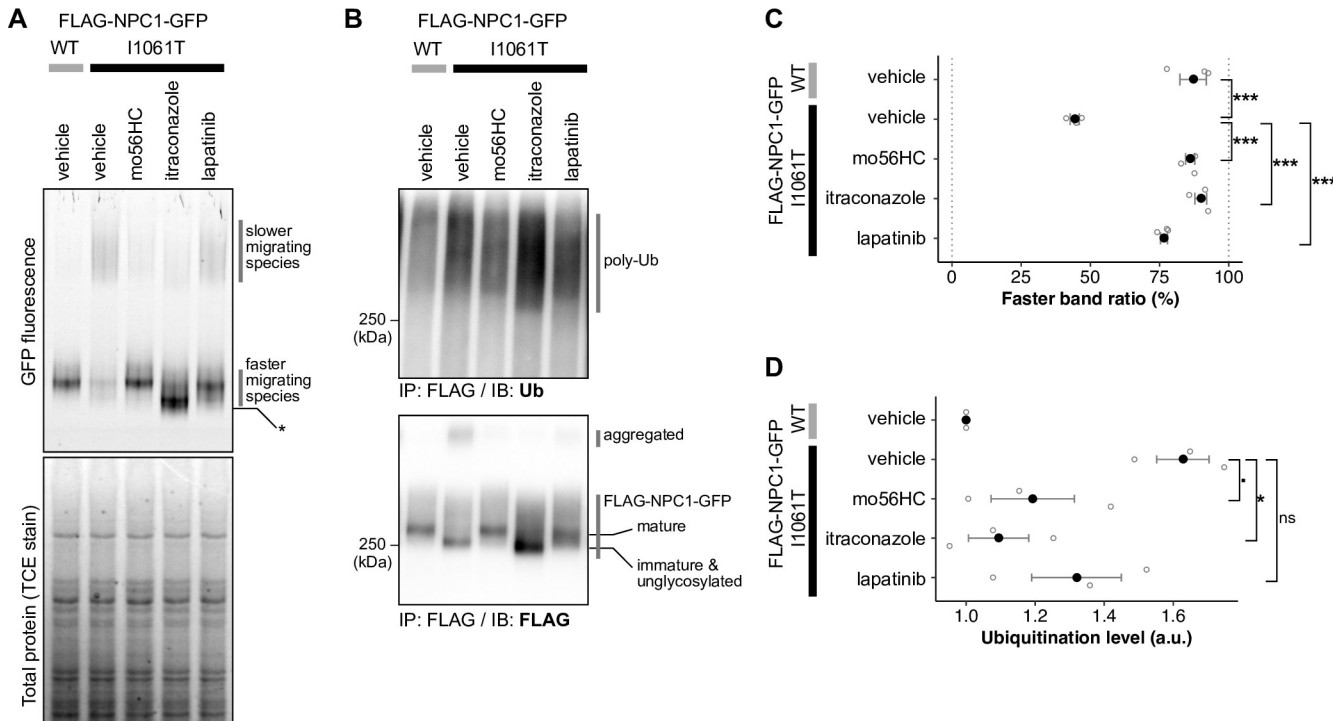

**Fig 4. Effect of itraconazole and lapatinib on the hydrodynamic status on native-PAGE gels and on the ubiquitination status.** (A) High-resolution clear native PAGE (fluorescence-detectable native PAGE) analysis of NPC1-GFP from cells treated as indicated. Cells stably expressing the indicated NPC1-GFP were treated with vehicle, mo56HC (3 µM), itraconazole (3 µM), or lapatinib (10 µM) for 21 h, and DDM-solubilized lysates were subjected to native-PAGE analysis after normalization with respect to total protein concentration. The native-PAGE gel was imaged for GFP fluorescence (upper panel), and total protein was also visualized by TCE staining (lower panel). (B) Ubiquitination status of NPC1-GFPs upon treatment with itraconazole or lapatinib. Cells stably expressing the indicated NPC1-GFP were treated as shown for 18 h followed by treatment with CB5083 (3 µM) for 6 h. The ubiquitination status was first probed with anti-Ub antibody after immunoprecipitation of the FLAG-NPC1-GFP, and then the amount of the FLAG-NPC1-GFP was re-probed with anti-FLAG antibody. The extent of ubiquitination was quantified and normalized to the amount of detected FLAG-NPC1-GFP. The raw, uncropped blot images are available from Mendeley Data repository (http://dx.doi.org/10.17632/jr23ccpp46.3). (C) Quantification of the high-resolution clear native PAGE analysis from three independent experiments. The filled circles, error bars, and open circles represent mean, SEM, and raw data points from three independent experiments. Statistical significance was assessed by ANOVA and Tukey-Kramer multiple comparison test (***, p<0.001). The raw, uncropped gel images are available from Mendeley Data repository (http://dx.doi.org/10.17632/jr23ccpp46.3). (D) Quantification of the ubiquitination assay data collected from three independent experiments. Data was normalized with the ubiquitination level of WT for each experiment. The filled circles, error bars, and open circles represent mean, SEM, and raw data points. Statistical significance of the treatment was assessed similarly as in (C). *, p<0.05; p<0.1; ns, not significant (p = 0.24).

former condition for routine use due to the instability of mCBB [39]. CN-PAGE analyses revealed different electrophoretic patterns for wild-type NPC1-GFP and NPC1-I1061T-GFP (**Fig 4A** and **4C**). While wild-type NPC1-GFP migrated mainly as the faster migrating band, less of this faster band was observed for the I1061T mutant and a significant portion of the mutant protein migrated as slower migrating, high molecular-weight (MW) species. Please note that the CN-PAGE is reported to give larger apparent molecular weight for membrane proteins compared to the soluble molecular weight markers, due to bound detergent micelles, and we could not deduce the size of the observed species [26]. Considering that thermal denaturation of proteins results in a reduced monodisperse band/peak [40, 41], a plausible explanation would be that the slower migrating band is possibly arisen from oligomerization due to partial denaturation and the faster/slower band ratio reflects the stability and/or folding status of WT and the I1061T mutant. Consistent with this idea, we observed a clear increase of the faster band upon treatment of the mutant with mo56HC or itraconazole (**Fig 4C**). Lapatinib also exhibited a similar effect, albeit less potent than mo56HC or itraconazole. These data

suggest that itraconazole, posaconazole, and lapatinib increase the stability of the mutant protein, probably by enhancing the folding efficiency, in a similar manner to mo56HC. Digoxin was again confirmed to reduce the NPC1-GFP expression level, but a slight increase in the faster/slower ratio was observed, which may imply slightly improved proteostasis of the NPC1 mutant.

Notably, itraconazole treatment resulted in a monomer band of higher electrophoretic mobility. As this effect was not observed when itraconazole was added after NPC1 solubilization, the difference is likely due to some perturbation of NPC1 protein synthesis. The most likely explanation would be the previously reported inhibitory effect of itraconazole on N-glycoside processing [42].

## Itraconazole and lapatinib reduce ubiquitination of the NPC1-I1061T mutant

Next, we examined the effect of itraconazole on the ubiquitination status of NPC1-I1061T. The I1061T mutant protein is primarily degraded through the ubiquitin-proteasome pathway [13, 14], and chaperone compounds should reduce ubiquitination of the mutant by helping it to escape the ER quality control machinery. So, we examined whether treatment with the hit compounds could reduce the extent of ubiquitination of the I1061T mutant. Consistent with the stabilizing effect of itraconazole, lapatinib, and mo56HC, these compounds also clearly reduced the ubiquitination level relative to the I1061T protein level (**Fig 4B** and **4D**).

## Design & synthesis of chemical probes based on the hit compounds

So far, our data supported the idea that itraconazole and lapatinib serve as pharmacological chaperones for NPC1-I1061T. As pharmacological chaperones exert their stabilizing effect via direct binding to the mutant protein, we next aimed to confirm direct binding of the compounds to NPC1-I1061T by means of photoaffinity crosslinking. For this purpose, we designed and synthesized photo-crosslinking probes based on itraconazole and lapatinib (**Fig 5A**; see **S1 Appendix** for details of the synthesis of the probes). We confirmed that the synthesized probes retained their activity towards the NPC1 mutant (**Fig 5B**). ItraAZY, a minimally modified photo-crosslinking probe with aryl azide as the photoreactive group and an alkyne moiety as a click chemistry handle [20, 43], was found to fully retain its activity towards the NPC1 mutant. In contrast, ItraACT, which was designed using the recently developed photo-crosslinker aryl-5-carboxytetrazole [44], was completely inactive. We also designed a pull-down probe, itra-BIO, for pull-down binding assay [45], but this probe was barely active. Unfortunately, lapaAZY, a photo-crosslinking analogue of lapatinib, showed no detectable activity in our assays, possibly in part due to the weak activity of the parent compound itself. So, we focused on itraAZY for the following experiments.

## Itraconazole directly binds to NPC1-I1061T competitively with respect to oxysterol-based chaperones

Using the highly active photo-crosslinking probe itraAZY, we performed photoaffinity labeling of FLAG-tagged NPC1-I1061T-GFP according to a protocol similar to the one we previously described [20]. Briefly, membrane preparation from HEK293 cells stably expressing FLAG-NPC1-I1061T-GFP was treated with itraAZY (and competitors) and irradiated with UV for 20 minute on ice. The irradiated membrane was solubilized, and a biotin-azide was introduced onto the alkyne moiety of the itraAZY via copper-catalyzed azide-alkyne cycloaddition reaction [30, 46]. After immunoprecipitation with anti-FLAG beads, itraAZY-labeled

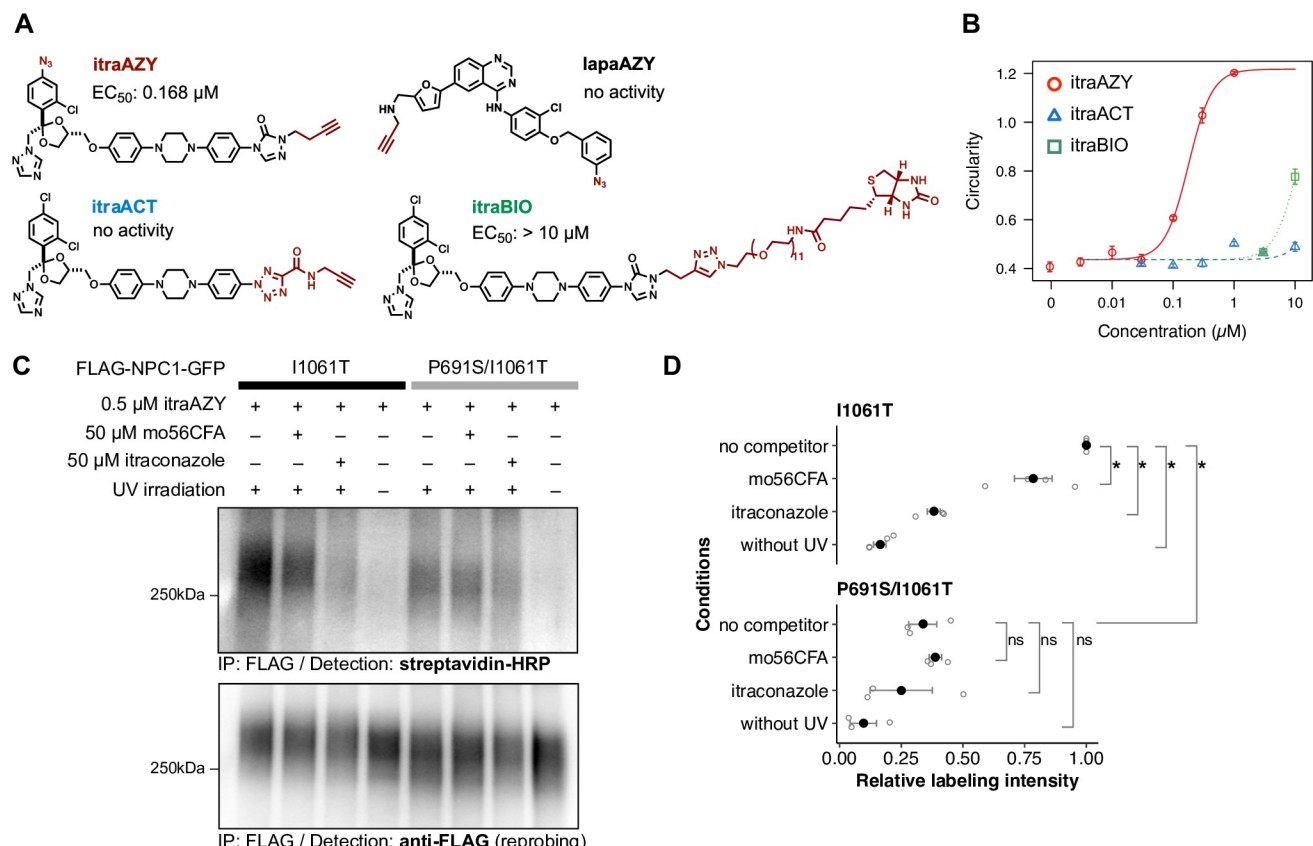

**Fig 5. Direct binding of itraconazole to NPC1-I1061T mutant protein.** (A) Structures of photoaffinity probes and pull-down probes for itraconazole and lapatinib. (B) Chaperone activity of the probes on NPC1-I1061T mutant protein, examined by image-based chaperone assay. (C) Photoaffinity labeling of FLAG-tagged NPC1-I1061T-GFP or NPC1-P691S/I1061T-GFP with itraAZY probe. Membrane fractions were prepared from the cells stably expressing the indicated NPC1-GFP, and photo-crosslinking was performed in the presence or absence of the indicated competitors. After conjugation of biotin to the alkyne via click chemistry, the NPC1 was immunoprecipitated and the presence of biotin was probed with streptavidin-HRP conjugate, followed by re-probing with anti-FLAG antibody. The raw, uncropped blot images are available from Mendeley Data repository (http://dx.doi.org/10. 17632/jr23ccpp46.3). (D) Quantification of the labeling. The extent of labeling was quantified and normalized with respect to the amount of the FLAG-NPC1-GFPs. Data is from independent experiments (n = 4 for I1061T and n = 3 for P691S/I1061T), and expressed as relative labeling respect to I1061T labeled with itraAZY. The filled circles, open circles, and error bars represent mean, raw data, and SEM, respectively. Statistical significance was assessed by the exact Wilcoxon rank sum test with correction of multiple comparisons by Benjamini-Hochberg method (*, p = 0.05; ns, not significant).

NPC1 mutant was quantitated. As shown in **Fig 5C and 5D**, itraAZY crosslinked the NPC1-I1061T mutant in a UV-irradiation-dependent manner, and competition with excess itraconazole or mo56CFA attenuated the crosslinking. This result indicates that itraconazole directly binds to NPC1-I1061T, in accordance with the reported interaction of itraconazole or posaconazole with WT NPC1 [47, 48], at the same site, or an adjacent or overlapping site, as the sterols. These data support our hypothesis that itraconazole directly binds to NPC1-I1061T, acting as a pharmacological chaperone to enhance the folding and stability of the mutant protein.

## Binding of itraconazole and oxysterol-based chaperones is sensitive to P691S mutation in the sterol-sensing domain

A mutation in the sterol-sensing domain (SSD), P691S (**Fig 1A**), interferes with the interaction of NPC1 with small molecules, including cholesterol [49], oxysterol-based chaperones [50], U18666A [51], posaconazole [48], and itraconazole [47]. Consistent with those reports, the

extent of labeling with itraAZY was significantly reduced in P691S/I1061T double mutant, implying that P691S mutation directly or indirectly interferes with itraconazole binding to NPC1-I1061T (**Fig 5C** and **5D**), as seen with the sterols.

We previously showed that the oxysterol-based chaperones exert their effect through sterol-binding site(s) other than the N-terminal domain (NTD), a well-characterized sterol-binding domain of NPC1 [52–55]. To see if the azoles also act independently of NTD, we tested the chaperone activity of itraAZY on NTD-deleted NPC1-GFP harboring the I1061T mutation (ΔNTD-I1061T) [20] or P691S/I1061T double mutations (ΔNTD-P691S/I1061T) [50]. For ΔNTD-I1061T, both the oxysterol-derived chaperone (mo56AZK) and the azole (itraAZY) clearly corrected the ER-localization to LAMP1-positive LE/L localization (see **Fig 6A** for representative images and **Fig 6B** for colocalization analysis). This result indicates that NTD is not required for the chaperone activity, again implying that the oxysterol-derived chaperone and the azole work in similar ways. Consistent with the sensitivity of itraAZY binding to the P691S mutation, the ΔNTD-P691S/I1061T mutant was refractory to both mo56AZK and itraAZY, confirming the importance of the intact sterol-sensing domain.

## Effect of itraconazole on cholesterol accumulation in NPC patient-derived fibroblasts

Finally, we examined whether itraconazole could reduce the accumulation of free cholesterol in the LE/L compartment of fibroblasts harboring I1061T mutation on both alleles, by means of filipin staining (**Fig 7**) [56]. As previously demonstrated [20], mo56HC treatment reduced cholesterol accumulation at 1 μM, while at 10 μM apparent inhibition of the NPC1 protein was observed; such a biphasic response is commonly observed with pharmacological chaperones [18, 57]. Itraconazole treatment, however, reduced cholesterol accumulation much less efficiently than mo56HC treatment, and only a slight reduction was only observed at 10 μM. Thus, although itraconazole was slightly more potent than mo56HC in correcting the folding defect of the I1061T mutant, it seemed to be less efficient for the functional rescue of NPC1-I1061T, for reasons that are currently unclear.

## Discussion

Since a folding defect of NPC1 plays a critical role in the etiology of Niemann-Pick disease type C [13], efforts have been made to identify drug candidates that can improve the folding efficiency of NPC1 mutant proteins. Such efforts led to the identification of proteostasis regulators, including a ryanodine receptor antagonist and HDAC inhibitors [58–61], and pharmacological chaperones specific to NPC1 [20–22]. As a part of our continuing program to find pharmacological chaperones for NPC1, in this work we have identified a subset of azole antifungals, including itraconazole, and a kinase inhibitor as potential chaperone drugs, by employing an image-based drug-repurposing screening system that monitors correction of mis-localization. Itraconazole showed an activity profile similar to that of our oxysterol-based chaperone, and photo-crosslinking experiments suggested both compounds bind to NPC1-I1061T at the same or an adjacent binding site, supporting our hypothesis that itraconazole acts as a pharmacological chaperone for NPC1. However, in contrast to the functional rescue achieved with the oxysterol chaperones, itraconazole was much less efficient in reducing cholesterol accumulation in patient-derived fibroblasts, suggesting that itraconazole itself may not be suitable for treatment of Niemann-Pick disease type C. Still, considering its high potency in correcting the localization defect of the NPC1 mutant, it could be a promising lead compound for the development of non-steroidal chaperone drugs through further medicinal chemistry optimization.

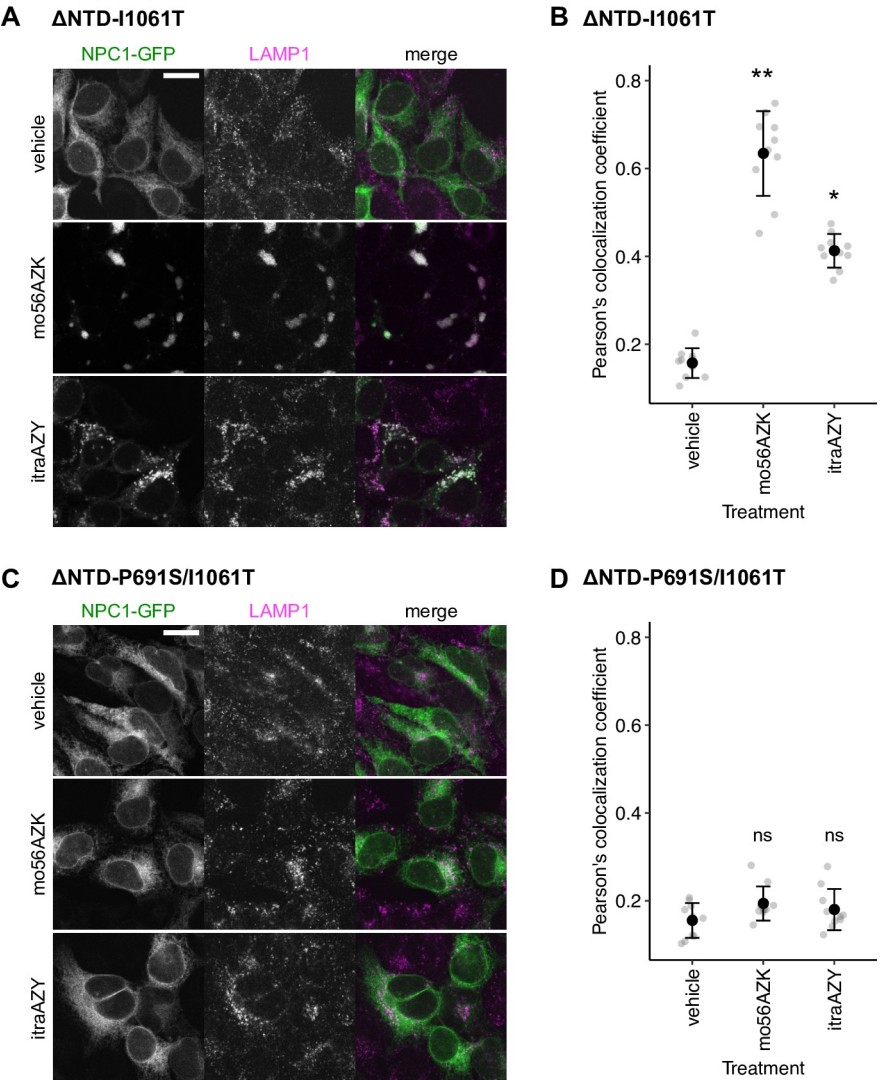

**Fig 6. Chaperone effect of mo56HC and itraconazole is not affected by deletion of NTD but is abrogated by P691S mutation.** (A) Colocalization of ΔNTD-NPC1-I1061T-GFP and LAMP1. Cells stably expressing ΔNTD-NPC1-I1061T-GFP were treated as indicated (1 μM mo56AZK or 3 μM itraAZY for 21 h). After LAMP1 immunostaining, images were acquired on a confocal microscope FV-3000 (Olympus) equipped with x60 objective lens. Scale bar, 50 μm. (B) Quantitative colocalization analysis of ΔNTD-NPC1-I1061T-GFP and LAMP1. The filled black circles and error bars denote mean ± SD from 10 images, and gray points represent raw correlation coefficients obtained for each image. Statistical significance was evaluated by applying the Kruskal-Wallis rank sum test along with Dunn's multiple comparison with the Benjamini-Hochberg-adjusted p value presented as *p<0.05 and **p<0.01 (two-sided). (C) Colocalization of ΔNTD-NPC1-P691S/I1061T-GFP and LAMP1, performed as in (A). (D) Quantitative colocalization analysis of ΔNTD-NPC1-P691S/I1061T-GFP and LAMP1 performed as in (B).

During the course of our study [62], itraconazole and posaconazole have been reported to inhibit NPC1 function via direct binding to the protein, leading to intracellular cholesterol accumulation [47, 48], which we confirmed by filipin staining of WT fibroblasts (**S5 Fig**). Such inhibitory activity may partly explain the failure to rescue cholesterol accumulation in our study, but the lack of a biphasic response, a common observation for pharmacological chaperones derived from inhibitors or antagonists, argues against this possibility. A previous report on the inhibitory effect of itraconazole on EGFR glycosylation [42] suggested another

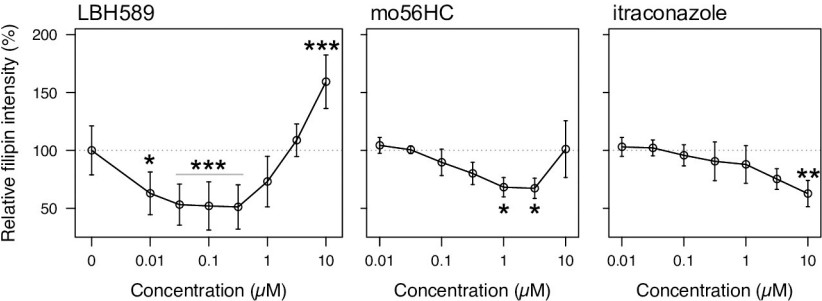

**Fig 7. Effect of itraconazole on cholesterol accumulation in NPC fibroblasts bearing I1061T mutation.** NPC fibroblasts (GM18453, $NPC1^{I1061T/I1061T}$) were treated with the indicated compounds for 48 h and cholesterol accumulation was evaluated by means of filipin staining. The extent of cholesterol accumulation was quantified as fractional filipin intensity within filipin-positive puncta over total filipin staining intensity within cells. Statistical significance was evaluated by applying Dunnett's multiple comparison test (two-sided) with adjusted p value (single-step method) presented as $^*p < 0.05$, $^{**}p < 0.01$, and $^{***}p < 0.001$ versus vehicle, after confirming normality and variance homogeneity by means of the Shapiro-Wilk test and Bartlett test. Error bars represent means ± SD from n = 3 (for LBH589) or n = 4 (vehicle, mo56HC, and itraconazole) independent experiments.

possibility, namely that itraconazole inhibited glycosylation of NPC1, leading to the production of non-functional NPC1 protein. This may also explain the observed amelioration of cholesterol accumulation at higher concentration. Considering that heavily glycosylated membrane proteins line the inner LE/L membrane to form a physical barrier known as the glycocalyx, and that global inhibition of glycosylation could potentially reduce the thickness of the glycocalyx, leading to an increase of NPC1-independent efflux of cholesterol from the compartment [63], such inhibition of glycocalyx synthesis may explain the observed slight reduction of cholesterol accumulation. Thus, our data may point to the possible utility of itraconazole, an orally available drug, as a glycocalyx synthesis inhibitor to reduce cholesterol accumulation in the LE/L compartment. Obviously, further studies would be needed to test this idea and to clarify the mechanism underlying the inhibition of glycosylation by itraconazole.

Contrary to the previous notion that NPC1 is degraded primarily through ubiquitin-proteasome system, recent study proposed that significant portion of NPC1-I1061T is also degraded through selective ER autophagy (ER-phagy) pathway [64, 65]. NPC1-I1061T entering ER-phagy pathway is delivered to lysosome for degradation, in a non-functional, immature form. Currently, we cannot exclude a possibility that itraconazole-treated NPC1-I1061T escapes degradation through ERAD but still selected for ER-phagy, making the non-functional mutant protein localize in the lysosomes. Further studies would be needed to clarify the contribution of ER-phagy in the itraconazole-mediated stabilization and alteration of NPC1-I1061T proteostasis.

Previously we have proposed the existence of a non-NTD, second sterol-binding site(s) on NPC1 that is important for chaperone activity [20]. Here we showed that both the oxysterol-derived chaperone and itraconazole bind to the putative non-NTD binding site(s) in a competitive manner and that the binding is sensitive to P691S mutation in the SSD. However, the exact location of the non-NTD sterol-binding site(s) remains elusive. X-ray and cryo-EM structures of NPC1 have revealed the presence of a cavity on the SSD [66, 67], but no electron density due to sterol has been observed. Recently reported cryo-EM structures of Ptch1, an SSD-containing protein with cholesterol transport-like activity, revealed the presence of sterol-like electron densities in the SSD cavity and in the channel-like conduit between the MLD and CTD loops (Fig 1A) [68], and further biochemical and structural studies demonstrated the presence of cholesterol in a pocket between MLD and CTD [69]. These data suggest that

NPC1 may also have potential ligand/substrate-binding sites in the corresponding channel-like conduit in addition to the SSD cavity, and recent research on NCR1, an yeast ortholog of human NPC1 [8], supports this possibility. FTMap analysis [70], which can predict ligand-binding hot spots by using small organic molecules as probes, also indicated that the conduit is a potential hot spot on NPC1. Consistent with these considerations, a recent cryo-EM structure of NPC1 in complex with itraconazole confirmed the presence of the conduit and located the itraconazole-binding site at the core of the conduit [71]. Such information should be helpful in logically designing itraconazole derivatives as NPC1 ligands in the future. To obtain a clearer understanding of how pharmacological chaperones stabilize NPC1 mutants and how NPC1 transports cholesterol, structural elucidation of NPC1-sterol complexes will be an important next step.

Our screen also identified lapatinib as another probable chaperone for NPC1, albeit with weak potency. This class of compounds has not so far been associated with NPC1, and may serve as an alternative starting point for developing novel NPC1 chaperones through medicinal chemistry efforts. In addition, digoxin, a cardiac glycoside that enhances cardiac contractility by indirectly altering $Ca^{2+}$ availability through inhibition of $Na^+$, $K^+$-ATPase, was also found to partially correct mis-localization of the mutant protein. Our data indicate that digoxin is unlikely to act as a pharmacological chaperone, and the mechanism of its effect is not clear. The most likely explanation is that digoxin acts a proteostasis regulator, as previously reported for the correction of the folding-defective phenotype of a disease-causing mutant of cystic fibrosis transmembrane conductor (CFTR) by ouabain, another cardiac glycoside, which works through modulation of the $Ca^{2+}$ status [72]. However, in contrast to the case of CFTR, digoxin was highly toxic and failed to alleviate cholesterol accumulation under our experimental conditions.

Our finding of azoles and lapatinib as possible ligands for NPC1 raises an interesting question: do they also bind to and inhibit another cholesterol transporter, NPC1-like 1 (NPC1L1)? NPC1L1 plays an important role in intestinal cholesterol absorption [73], and is the target of ezetimibe, a cholesterol absorption inhibitor [74, 75]. Our previous studies have suggested that NPC1 and NPC1L1 show similar but distinct preferences in sterol binding [76, 77], so it seems possible that some azoles and/or lapatinib may also inhibit NPC1L1. Considering that sterol derivatives and ezetimibe do not share a binding site [76] and that fomiroid A [78], a steroid isolated from a mushroom, has been found to inhibit NPC1L1 via a distinct mode of action from that of ezetimibe, our findings could also be relevant to uncovering a new class of cholesterol absorption inhibitors. In addition, recent reports suggest that NPC1L1 is involved in vitamin K absorption and drug-drug interaction between ezetimibe and warfarin, a vitamin K antagonist [79], so the effect of the azoles and lapatinib on vitamin K absorption would also need further evaluation.

In summary, we have identified several classes of compound that improve the proteostasis of mutant NPC1 by the application of a newly developed image-based drug screening system. By employing a photoaffinity labeling approach, we established that itraconazole directly binds to the NPC1-I1061T mutant protein to help it escape ER retention and subsequent premature degradation by ERAD. Further, our data imply that itraconazole and oxysterol-derived chaperones, and possibly lapatinib, may share the same binding site, or adjacent binding sites, on the NPC1 protein. Although itraconazole itself failed to improve the loss-of-function phenotype of Niemann-Pick disease type C, we believe our findings offer good starting points for developing better chaperone drugs based on the chemical structures of itraconazole and lapatinib to treat this disease.

## Supporting information

**S1 Appendix. Supplementary materials and methods, including synthetic schemes, procedures, and characterization data for itraAZY, itraACT, itraBIO, and lapaAZY.**
(PDF)

**S1 Fig. Analytical workflow and representative intermediate results for the evaluation of cholesterol accumulation visualized by filipin staining.** (A) Schematic workflow of the analysis of filipin-stained images. (B) A concatenated intensity profile from randomly selected images, showing the presence of high-intensity filipin-positive regions (mostly LE/L compartment), low-intensity regions originated from within-cell area, and background regions originated from areas without cells. The red lines on the right of the plot represent cluster means, and the blue lines represent the "lower threshold" and "higher threshold". Note that image intensity in R is scaled to 0 to 1. (C) Distribution of the pixel intensities plotted in (B), along with the lower and higher thresholds, showing the presence of background clusters. (D) Representative intermediate images from the workflow, showing successful extraction of cell-covered area and filipin-positive, vesicular area.
(JPG)

**S2 Fig. Example images obtained from the screening.** Representative screening images from cells treated with mo56HC, itraconazole, lapatinib, bortezomib, and digoxin are shown. To better visualize the subcellular localization pattern, contrast of each images was adjusted image-by-image basis. Scale bar, 50 μm.
(PDF)

**S3 Fig. Viability assay of the selected hit compounds.** HEK293 cells were treated with the indicated concentrations of itraconazole, digoxin, lapatinib, and bortezomib for 20 h, and viability of the cells was assessed by alamarBlue assay. The data points represent the mean from three independent experiments each performed in triplicate, and the error bars denote standard deviations from the independent experiments. The $IC_{50}$ values shown in the table were calculated from each independent experiment, and represented as mean ± SD (n = 3).
(PDF)

**S4 Fig. Up-regulatory effect of mo56HC and itraconazole on NPC1-I1061T mutant in the presence of a protein synthesis inhibitor, emetine.** To confirm that the up-regulatory effect of itraconazole is due to reduced degradation of the NPC1-I1061T-GFP, we tested if itraconazole shows up-regulatory effect even in the presence of emetine. The cells expressing NPC1-I1061T-GFP were treated as indicated in the presence of 25 μM emetine for 18 h, and the expression level of NPC1-I1061T-GFP was assessed as previously described (GFP measurement after lysis with TNET buffer, and normalized with total protein concentration measured by BCA assay) [20]. The data were independently collected three times, and each experiment was performed in biological triplicates. The open circles represent the biological replicates and the filled circles represent their average, and the data was visualized as a SuperPlot to better represent experimental variation and reproducibility [36]. Mean and SEM of the independent experiments were shown as bold lines and error bars. Statistical significance was assessed by paired t-test with Benjamini-Hochberg p adjustment for multiple comparisons (., p<0.1; *, p<0.05).
(PDF)

**S5 Fig. Effect of mo56HC and itraconazole on cholesterol accumulation in fibroblast with NPC1-WT (NPC fibroblast GM05659H) and NPC1-I1061T mutant (NPC fibroblast GM18453).** As reported previously [20], mo56HC treatment of NPC1-I1061T cells showed

biphasic response, where cholesterol accumulation is alleviated at lower concentration, but higher concentration of mo56HC rather induced cholesterol accumulation, which was also observed for WT cells. In contrast, itraconazole induced cholesterol accumulation in WT cells, but slightly reduced cholesterol accumulation in I1061T cells through currently unclear mechanisms. The data represents mean ± SD of nine fields of images.
(PDF)

## Acknowledgments

We thank the University of Tokyo IQB Olympus Bioimaging Center (TOBIC) for technical assistance and use of an FV3000 confocal microscope. The following cell line was obtained from the NIGMS Human Genetic Cell Repository at the Coriell Institute for Medical Research: GM18453 (NPC1 I1061T/I1061T) and GM05659H (apparently healthy fibroblast). We thank Professor Koji Kuramochi for his kind help during the revision of the manuscript.

## Author Contributions

**Conceptualization:** Yuichi Hashimoto, Mikiko Sodeoka, Kenji Ohgane.

**Data curation:** Ryuta Shioi, Fumika Karaki, Hiromasa Yoshioka, Kenji Ohgane.

**Formal analysis:** Ryuta Shioi, Fumika Karaki, Hiromasa Yoshioka, Kenji Ohgane.

**Funding acquisition:** Yuichi Hashimoto, Kenji Ohgane.

**Investigation:** Ryuta Shioi, Fumika Karaki, Hiromasa Yoshioka, Kenji Ohgane.

**Methodology:** Ryuta Shioi, Fumika Karaki, Tomomi Noguchi-Yachide, Minoru Ishikawa, Kosuke Dodo, Kenji Ohgane.

**Project administration:** Tomomi Noguchi-Yachide, Minoru Ishikawa, Kosuke Dodo, Yuichi Hashimoto, Mikiko Sodeoka.

**Resources:** Tomomi Noguchi-Yachide, Minoru Ishikawa, Kosuke Dodo, Yuichi Hashimoto, Mikiko Sodeoka.

**Supervision:** Tomomi Noguchi-Yachide, Minoru Ishikawa, Kosuke Dodo, Yuichi Hashimoto, Mikiko Sodeoka, Kenji Ohgane.

**Validation:** Ryuta Shioi, Hiromasa Yoshioka, Tomomi Noguchi-Yachide, Minoru Ishikawa, Kosuke Dodo, Yuichi Hashimoto, Mikiko Sodeoka, Kenji Ohgane.

**Visualization:** Ryuta Shioi, Kenji Ohgane.

**Writing – original draft:** Ryuta Shioi, Hiromasa Yoshioka, Kenji Ohgane.

**Writing – review & editing:** Fumika Karaki, Hiromasa Yoshioka, Tomomi Noguchi-Yachide, Minoru Ishikawa, Kosuke Dodo, Yuichi Hashimoto, Mikiko Sodeoka, Kenji Ohgane.

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
