## [Decision Letter · Decision Letter 0]

17 Jun 2020

PONE-D-20-11428

Image-based screen capturing misfolding status of Niemann-Pick type C1 identifies potential candidates for chaperone drugs

PLOS ONE

Dear Dr. Ohgane,

Thank you for submitting your manuscript to PLOS ONE. After careful consideration, we feel that it has merit but does not fully meet PLOS ONE’s publication criteria as it currently stands. Therefore, we invite you to submit a revised version of the manuscript that addresses the points raised during the review process.

Specifically, both reviewers deemed the article interesting, but they raised significant concerns that have to be addressed before the manuscript becomes acceptable for publication. The paper will likely undergo a second round of review.

We look forward to receiving your revised manuscript.

Kind regards,

Oscar Millet

Academic Editor

PLOS ONE

Journal Requirements:

Reviewers' comments:

Reviewer's Responses to Questions

**Comments to the Author**

1. Is the manuscript technically sound, and do the data support the conclusions?

Reviewer #1: Partly

Reviewer #2: Yes

2. Has the statistical analysis been performed appropriately and rigorously? 

Reviewer #1: Yes

Reviewer #2: Yes

3. Have the authors made all data underlying the findings in their manuscript fully available?

Reviewer #1: Yes

Reviewer #2: Yes

4. Is the manuscript presented in an intelligible fashion and written in standard English?

Reviewer #1: Yes

Reviewer #2: Yes

5. Review Comments to the Author

Reviewer #1: Additional insights in the mechanism of drug action will further support the hypothesis. The authors also need to include images from the original screen to increase the readers understanding of what hits vs. non-hits look like.

Reviewer #2: This is a thorough and extensive work of Shioi et al, who seek to find repositioned drugs as pharmacological chaperones for lysosomal cholesterol transporter NPC1, using an image-based cellular assay. Methods and drug screening procedures are extensively reported, and hits resulted from the screening further analysed in fibroblasts carrying a mutated NPC1.

Although the manuscript has potential to be accepted on Plos One journal, in the reviewer's opinion, additional experiments/comments should be addressed.

1) Major points.

One important hit obtained in the screening, itraconazole, shows a positive effect in the filipin assay (fig 7), although milder than mo56HC. I see that the point at 5 micromolar has a milder but maybe functionally relevant effect. If it was repeated 1-2 times more it could reach significance. This drug may have more therapeutic interest than mo56HC, as higher doses don't have an opposite effect on the assay.

Another important point is that itraconazole has an opposite role on wild type NPC1 protein. The authors comment this effect on discussion section, and give reference 66. However, I think there is not enough explanation on this, nor its potential functional relevance, and even there are more studies reporting NPC1 inhibition by itraconazole (Trinh et al 2017, Head et al 2017). Indeed, what would happen on the filipin assay with WT fibroblasts treated with itraconzaole? Would it reach same levels as with mutated NPC1, or lower ones? A filipin assay with WT protein and more discussion on the matter could clarify this point.

2) Minor points. Please revise general text for typos. e.g. Laemmli instead of Laemli, thousands without commas (line 217, 2000xg vs 125,000xg), etc.

6. PLOS authors have the option to publish the peer review history of their article (what does this mean?). If published, this will include your full peer review and any attached files.

Reviewer #1: No

Reviewer #2: No

---

## [Author Response · Author response to Decision Letter 0]

14 Sep 2020

### For complete response (with formatting and inserted figures), please refer to the Response to reviewers file ###

Response to the editor’s & reviewers’ comments

Re: Editor’s comment

Thank you for submitting your manuscript to PLOS ONE. After careful consideration, we feel that it has merit but does not fully meet PLOS ONE’s publication criteria as it currently stands. Therefore, we invite you to submit a revised version of the manuscript that addresses the points raised during the review process.

Specifically, both reviewers deemed the article interesting, but they raised significant concerns that have to be addressed before the manuscript becomes acceptable for publication. The paper will likely undergo a second round of review. 

We are grateful to the Editor and reviewers for evaluating our manuscript in this difficult time, giving constructive feedback, and allowing us to revise the manuscript. We are also happy to hear that the reviewers share our enthusiasm for our findings.

 We have performed new experiments to address the concerns raised and revised the manuscript accordingly, as detailed in the following point-by-point responses. Most importantly, the revised manuscript now includes (1) representative images from the image-based screen, showing how the hit compounds alter the localization pattern of NPC1-I1061T-GFP (Fig S2), (2) an additional insight into how itraconazole works on NPC1-I1061T (upregulatory effect of itraconazole on NPC1-I1061T protein in the presence of a protein synthesis inhibitor) (Fig S4), (3) independent viability assay data of the hit compounds (Fig S3), (4) quantification of all the gel data along with their statistical analyses in response to the reviewer #1’s comment (revised Fig 4C, 4D, and 5D), and (5) the data of itraconazole effect on the cholesterol accumulation in WT cells in addition to the I1061T mutant cells in response to the reviewer #2’s suggestion (Fig S5).

Additionally, to make it clear that currently we cannot exclude possible contribution of ER autophagy pathway to the itraconazole’s effect on NPC1 proteostasis as suggested by the reviewer #1, we have added a paragraph discussing this matter. Also, typographical mistakes pointed out by the reviewer #2 were corrected, and raw, uncropped gel images, and data set used to make plots were uploaded to the Mendely Data repository to meet the PLOS One’s data policy.

 Thanks to the Editor and the reviewers support, we believe our manuscript has now been significantly improved. We hope that our revised manuscript now meets the PLOS ONE’s publication criteria.

 

Re: Reviewer #1

Additional insights in the mechanism of drug action will further support the hypothesis. The authors also need to include images from the original screen to increase the readers understanding of what hits vs. non-hits look like. 

We are grateful to the reviewer for providing constructive feedbacks detailed below. We added further discussions and included the results from additional experiments to gain more insights into the mechanisms and to support our hypothesis (see the following point-by-point responses). Also, we included representative images from the original screen in the supplementary Fig S2.

[from the attached, separate review file] Shioi et al develop an image-based screen to discover novel NPC1-I1061T chaperones. This is a timely study as chaperone-based therapeutics have become of significant interest in the Niemann-Pick C. Field. The authors use a GFP NPC1-I1061T fusion system to develop, implement a drug-repurposing screen, and interpret their positive hits. The authors obtain several promising leads and further characterize their effect of I1061T ubiquitination and function. This is exciting and novel work; however, the authors need to address the following point to confirm their hypothesis and better understand how these compounds are functioning on NPC1. 

We are happy to hear the reviewer’s enthusiasm and thank the reviewer for raising important points to understand the mechanism of action of the chaperone candidates.

1: The bortezomid and itraconazole screening images should be included in the manuscript for comparison. Currently it is only discussed. 

As suggested, we have added the representative images from the original screen in the supplementary Fig S2 (attached below). Although the screen was performed with a non-confocal cell imager (IN Cell Analyzer 2000) and therefore the resolution of the images were not so high, the change in the localization pattern of NPC1-I1061T-GFP could be easily recognized and quantified by image analysis. Itraconazole treatment resulted in a NPC1-I1061T-GFP pattern similar to that of mo56HC-treated cells. In contrast, bortezomib resulted in large aggregate-like structures distinct from the mo56HC-treated NPC1.

 

Supplementary Figure S2. Example images obtained from the screening. Representative screening images from cells treated with mo56HC, itraconazole, lapatinib, bortezomib, and digoxin are shown. To better visualize the subcellular localization pattern, contrast of each images was adjusted image-by-image basis. Scale bar, 50 µm.

 

2: All gels need to be quantified. 

The revised manuscript now includes quantified results from more than three independent experiments as new panels of figures (new Fig 4C for CN-PAGE quantification, new Fig 4D for ubiquitination assay, and new Fig 5D for photoaffinity labeling). 

Fig 4C. (C) Quantification of the high-resolution clear native PAGE analysis from three independent experiments. The filled circles, error bars, and open circles represent mean, SEM, and raw data points from three independent experiments. Statistical significance was assessed by ANOVA and Tukey-Kramer multiple comparison test (***, p<0.001).

Fig 4D. (D) Quantification of the ubiquitination assay data collected from three independent experiments. Data was normalized with the ubiquitination level of WT for each experiment. The filled circles, error bars, and open circles represent mean, SEM, and raw data points. Statistical significance of the treatment was assessed similarly as in (C). *, p<0.05; ., p<0.1; ns, not significant (p=0.24).

.

Fig 5D. (D) Quantification of the labeling. The extent of labeling was quantified and normalized with respect to the amount of the FLAG-NPC1-GFPs. Data is from independent experiments (n=4 for I1061T and n=3 for P691S/I1061T), and expressed as relative labeling respect to I1061T labeled with itraAZY. The filled circles, open circles, and error bars represent mean, raw data, and SEM, respectively. Statistical significance was assessed by the exact Wilcoxon rank sum test with correction of multiple comparisons by Benjamini-Hochberg method (*, p = 0.05; ns, not significant).

3: The authors need to better establish how their lead compound itraconazole modulates I1061T. The authors indicate itraconazole reduces ubiquitination however, 4B indicates an increase in poly-ub with itraconazole? 

As the reviewer noted, poly-Ub signal per se indeed increased upon itraconazole treatment. However, itraconazole also increased NPC1-I1061T protein level in the sample. Actually, the ratio of the poly-ubiquitination signal per NPC1-I1061T signal decreased slightly as shown in the quantification plot now included as new Fig 4D. The upregulation of NPC1-I1061T level by itraconazole is due to stabilization of the mutant protein as shown by flow cytometric analysis and western blotting, and our new experiments that confirmed reduced degradation of the mutant protein (see below for detail, minor point #2). To clarify this point, we slightly modified the main text accordingly as shown below (with underlines).

“… Consistent with the stabilizing effect of itraconazole, lapatinib, and mo56HC, these compounds also clearly reduced the ubiquitination level relative to the I1061T protein level (Fig 4B and D). …”

4: Shioi nicely show itraconazole increases co-localization of NPC1-I1061T with the lysosomal compartment and later show this lysosomal form does not correct filipin. Interestingly, fig. 4B bottom suggests that itraconazole accumulates an immature glycosylated form of NPC1-I1061T. The authors indicate itraconazole is known to inhibit glycosylation however, there is an alternative interpretation of this data. Schultz et al (Nature Comm 2018) showed NPC1-I1061T is also degraded from the ER by both ERAD and ER-autophagy. They similarly found that NPC1-I1061T from ER-phagy entering the lysosome was not-functional and EndoH sensitive. It is possible that itraconazole prevents ERAD but still allows ER- phagy to occur. The authors should perform an EndoH assay to empirically determine the glycosylation state of NPC1-WT and NPC1-I1061T plus and minus itraconazole or comment on this alternative interpretation. 

We thank the reviewer for the alternative interpretation of the immature glycosylation of the itraconazole-rescued NPC1-I1061T protein. We totally agree with the reviewer that we cannot rule out the contribution of the ER-phagy in the itraconazole-induced translocation of the NPC1-I1061T to lysosomes/late endosomes. Therefore, we discussed the possibility of the ER-phagy by adding the following paragraph in the discussion section.

“Contrary to the previous notion that NPC1 is degraded primarily through ubiquitin-proteasome system, recent study proposed that significant portion of NPC1-I1061T is also degraded through selective ER autophagy (ER-phagy) pathway [64, 65; Schultz et al. 2016 Brain Res & Schultz et al. 2018 Nat Commun]. NPC1-I1061T entering ER-phagy pathway is delivered to lysosome for degradation, in a non-functional, immature form. Currently, we cannot exclude a possibility that itraconazole-treated NPC1-I1061T escapes degradation through ERAD but still selected for ER-phagy, making the non-functional mutant protein localize in the lysosomes. Further studies would be needed to clarify the contribution of ER-phagy in the itraconazole-mediated stabilization and alteration of NPC1-I1061T proteostasis.”

Unfortunately, however, it would not be easy to experimentally dissect these two possibilities (ER-phagy-mediated localization to lysosomes v.s. chaperone-mediated translocation to lysosomes), as itraconazole globally inhibits normal glycosylation of membrane proteins, including EGFR, making them intrinsically EndoH sensitive. Thus, we think such investigation would be beyond the scope of the current manuscript, although the idea sounds interesting and important.

5: The authors should comment if there a pool of GFP-NPC1-I1061T which does not fold properly and cannot be seen in the clear-native PAGE gels. 

No recognizable level of NPC1-I1061T-GFP that did not entered the gel was seen in our experiment. We added the following brief comment in the Materials and Methods section of the CN-PAGE analysis.

“Note that no recognizable signal could be observed at the top of the gels under our experimental condition, implying that terminally misfolded NPC1 proteins were not soluble in the lysis buffer, if any.”

Minor 

1: Although the authors include nuclear staining to assess cell death, cell death needs to be formally assessed with dose curves of the lead compounds (at a minimum itraconazole). 

As suggested, we performed cell viability assay of the hit compounds by monitoring metabolic activity of viable cells with a resazurin-based reagent, alamarBlue. The cytotoxicity data is included in the supplementary Fig S3 (shown below), and briefly mentioned in the main text.

Supplementary Fig S3. Viability assay of the selected hit compounds. HEK293 cells were treated with the indicated concentrations of itraconazole, digoxin, lapatinib, and bortezomib for 20 h, and viability of the cells was assessed by alamarBlue assay. The data points represent the mean from three independent experiments each performed in triplicate, and the error bars denote standard deviations from the independent experiments. The IC50 values shown in the table were calculated from each independent experiment, and represented as mean ± SD (n=3).

2: The authors should determine if NPC1-I1061T stability is modified with itraconazole. 

Although half-life determination would be much better experiment on this regard, we could not perform such experiments due to restricted access to the lab facilities under the current COVID status in our university. Instead, we assessed the effect of itraconazole on the level of NPC1-I1061T in the presence of a protein synthesis inhibitor at one time point (there independent experiments were performed, and the result is shown in Supplementary Fig S4 below). Even in the presence of emetine, a protein synthesis inhibitor, itraconazole-mediated upregulation of NPC1-I1061T protein was observed, supporting our hypothesis that itraconazole decreases degradation rate of NPC1-I1061T, thereby increases its stability in the living cells.

Supplementary Fig S4. Up-regulatory effect of mo56HC and itraconazole on NPC1-I1061T mutant in the presence of a protein synthesis inhibitor, emetine. To confirm that the up-regulatory effect of itraconazole is due to reduced degradation of the NPC1-I1061T-GFP, we tested if itraconazole shows up-regulatory effect even in the presence of emetine. The cells expressing NPC1-I1061T-GFP were treated as indicated in the presence of 25 µM emetine for 18 h, and the expression level of NPC1-I1061T-GFP was assessed as previously described (GFP measurement after lysis with TNET buffer, and normalized with total protein concentration measured by BCA assay) [20; Ohgane et al. 2013 Chem. Biol.]. The data were independently collected three times, and each experiment was performed in biological triplicates. The open circles represent the biological replicates and the filled circles represent their average, and the data was visualized as a SuperPlot to better represent experimental variation and reproducibility [36; Lord SJ, Velle KB, Mullins RD, Fritz-Laylin LK. SuperPlots: Communicating reproducibility and variability in cell biology. J Cell Biol. 2020;219(6):94. doi:10.1083/jcb.202001064.]. Mean and SEM of the independent experiments were shown as bold lines and error bars. Statistical significance was assessed by paired t-test with Benjamini-Hochberg p adjustment for multiple comparisons (., p<0.1; *, p<0.05).

Re: Reviewer #2

This is a thorough and extensive work of Shioi et al, who seek to find repositioned drugs as pharmacological chaperones for lysosomal cholesterol transporter NPC1, using an image-based cellular assay. Methods and drug screening procedures are extensively reported, and hits resulted from the screening further analysed in fibroblasts carrying a mutated NPC1. 

Although the manuscript has potential to be accepted on Plos One journal, in the reviewer's opinion, additional experiments/comments should be addressed.

We thank the reviewer for sharing enthusiasm with us.

1) Major points.

One important hit obtained in the screening, itraconazole, shows a positive effect in the filipin assay (fig 7), although milder than mo56HC. I see that the point at 5 micromolar has a milder but maybe functionally relevant effect. If it was repeated 1-2 times more it could reach significance. This drug may have more therapeutic interest than mo56HC, as higher doses don't have an opposite effect on the assay. 

We agree with the reviewer in that the observed effect of itraconazole may be functionally relevant. However, unfortunately, we could not afford repeating the Filipin staining assay of NPC fibroblasts under our current situation (inability to access the cell imager used for the assay due to COVID, and unavailability of NPC fibroblasts in my current institute for now as I have just moved to another university). Still, as the reviewer suggested, the itraconazole’s effect on the cholesterol level is indeed interesting considering its pleiotropic activity, and we hope to gain more insight into the phenomenon in future.

Another important point is that itraconazole has an opposite role on wild type NPC1 protein. The authors comment this effect on discussion section, and give reference 66. However, I think there is not enough explanation on this, nor its potential functional relevance, and even there are more studies reporting NPC1 inhibition by itraconazole (Trinh et al 2017, Head et al 2017). Indeed, what would happen on the filipin assay with WT fibroblasts treated with itraconzaole? Would it reach same levels as with mutated NPC1, or lower ones? A filipin assay with WT protein and more discussion on the matter could clarify this point. 

The reports by Trinh et al. (2017) and Head et al. (2017), reporting the direct inhibition of NPC1 function by posaconazole and itraconazole, had already been cited in other paragraphs and discussed (current reference number 47 & 48). To answer the reviewer’s question “what would happen on the filipin assay with WT fibroblasts”, we added a figure comparing the effect of itraconazole and mo56HC on the cholesterol accumulation in both WT and I1061T-NPC1 cells (Supplementary Fig S5). The effect of mo56HC on WT and I1061T cells were straightforward, with the WT cells only increased filipin staining was observed at higher concentration. However, the interpretation of itraconazole effect is not straightforward; itraconazole increased cholesterol accumulation in WT cells but decreased that in I1061T cells at 10 µM. Actually, itraconazole treatment of WT and I1061T cells resulted in similar level of filipin staining intensity at 10 µM as predicted by the reviewer. However, considering that itraconazole is not well soluble at more than 10 µM and we are not sure if the itraconazole’s effect reached plateau with respect to its dose. Therefore, we think this “why” issue is beyond the scope of the current manuscript.

Thus, although we could not provide clear explanation on this phenomenon, we briefly mentioned the WT data (Fig S5) in the discussion as a remaining, unsolved issue.

2) Minor points. Please revise general text for typos. e.g. Laemmli instead of Laemli, thousands without commas (line 217, 2000xg vs 125,000xg), etc. 

We thank the reviewer for pointing this out. We have corrected the typos like the lack of commas.

---

## [Decision Letter · Decision Letter 1]

28 Sep 2020

PONE-D-20-11428R1

Image-based screen capturing misfolding status of Niemann-Pick type C1 identifies potential candidates for chaperone drugs

PLOS ONE

Dear Dr. Ohgane,

Thank you for submitting your manuscript to PLOS ONE. After careful consideration, we feel that it has merit but does not fully meet PLOS ONE’s publication criteria as it currently stands. Therefore, we invite you to submit a revised version of the manuscript that addresses the points raised during the review process.

There are a few number of remaining issues that need to be addressed before we can further consider the manuscript.

We look forward to receiving your revised manuscript.

Kind regards,

Oscar Millet

Academic Editor

PLOS ONE

Reviewers' comments:

Reviewer's Responses to Questions

**Comments to the Author**

1. If the authors have adequately addressed your comments raised in a previous round of review and you feel that this manuscript is now acceptable for publication, you may indicate that here to bypass the “Comments to the Author” section, enter your conflict of interest statement in the “Confidential to Editor” section, and submit your "Accept" recommendation.

Reviewer #1: (No Response)

2. Is the manuscript technically sound, and do the data support the conclusions?

Reviewer #1: Partly

3. Has the statistical analysis been performed appropriately and rigorously? 

Reviewer #1: Yes

4. Have the authors made all data underlying the findings in their manuscript fully available?

Reviewer #1: Yes

5. Is the manuscript presented in an intelligible fashion and written in standard English?

Reviewer #1: Yes

6. Review Comments to the Author

Reviewer #1: Considering the difficulties of performing laboratory work during COVID-19, Shioi et. al. did a commendable job addressing my major concerns. The following minor points still exist:

1) In fig 4A the authors describe I1061T as a monomer and oligomer. While NPC1 has been reported to be a dimer, this is still controversial and the authors focus on correcting the “monomeric” NPC1. The authors should include molecular weight markers on the gel so that we can at least see if the “oligomers” represent ubiquitinated NPC1 species. To simplify the nomenclature, the authors should consider renaming the monomer and oligomer to just NPC1 and high molecular weight species.

2) In the discussion the authors imply that since intraconazole binds a similar region as oxysterols it “establishes intraconazole as a pharmacological chaperone”. However, without proper I1061T glycosylation and correction of filipin this is not the case. The authors need to revise this statement to less definitive like they did in the abstract.

7. PLOS authors have the option to publish the peer review history of their article (what does this mean?). If published, this will include your full peer review and any attached files.

Reviewer #1: No

---

## [Author Response · Author response to Decision Letter 1]

12 Nov 2020

[Reviewer #1] Considering the difficulties of performing laboratory work during COVID-19, Shioi et. al. did a commendable job addressing my major concerns. The following minor points still exist:

>> We are really grateful to the reviewer for the kind consideration on the current situation.

[Reviewer #1] 1) In fig 4A the authors describe I1061T as a monomer and oligomer. While NPC1 has been reported to be a dimer, this is still controversial and the authors focus on correcting the “monomeric” NPC1. The authors should include molecular weight markers on the gel so that we can at least see if the “oligomers” represent ubiquitinated NPC1 species. To simplify the nomenclature, the authors should consider renaming the monomer and oligomer to just NPC1 and high molecular weight species.

>> We agree that the native oligomeric state of NPC1 remains a controversial; some reported as dimer and some as monomer.

First of all, we would like to clarify that the native PAGE protocol we used here (fluorescence-detectable, clear-native PAGE) utilizes a mixture of a mild anionic detergent and non-ionic detergent [ref 26 in the manuscript, Ihara et al., 2011 (Anal Biochem 412, 217-223)]. In contrast to the most-common blue-native PAGE, this method has been reported to give larger apparent molecular weight for membrane proteins compared to soluble proteins due to the bound detergent micelles [ref 26]. Thus, the commercially available molecular weight markers could not be used to deduce the actual size of the observed NPC1-GFP species, and indeed the “monomer” band was found near the 440 kDa marker (ferritine) and the “oligomer” band was observed slightly above the 669 kDa marker (thyroglobulin) in our preliminary experiments. Thus, to discuss the exact size of the observed species, we may need to use blue native-PAGE combined with western blotting analysis (as the blue color interferes with GFP detection), which we could not perform under the current situation.

So to clarify why we do not discuss the molecular weight of the observed species, we added a sentence “Please note that the CN-PAGE is reported to give larger apparent molecular weight for membrane proteins compared to the soluble molecular weight markers, due to bound detergent micelles, and we could not deduce the size of the observed species [26]”.

Regarding the second issue, “monomer/oligomer” issue, we would like to clarify that our main focus here is to show the different mobility of the NPC1-GFP bands and not to discuss the exact identity of the observed bands nor exact oligomeric status of the NPC1 GFPs. Therefore, following the suggestion by the reviewer, we renamed the “monomer/oligomer” to “faster migrating band/slower migrating species” to emphasize the electrophoretic difference rather than assuming their oligomerization state. However, we commented a possible identity of the slower migrating species as “possibly arisen from oligomerization due to partial denaturation” as in the following sentence: “Considering that thermal denaturation of proteins results in a reduced monodisperse band/peak [40,41], a plausible explanation would be that the slower migrating band is possibly arisen from oligomerization due to partial denaturation and the faster/slower band ratio reflects the stability and/or folding status of WT and the I1061T mutant.”

 Accordingly, the “monomer/oligomer” in the Fig 4A and the “monomeric peak ratio”in Fig 4C were replaced with “faster migrating species / slower migrating species” and “faster band ratio”.

[Reviewer #1] 2) In the discussion the authors imply that since intraconazole binds a similar region as oxysterols it “establishes intraconazole as a pharmacological chaperone”. However, without proper I1061T glycosylation and correction of filipin this is not the case. The authors need to revise this statement to less definitive like they did in the abstract. 

>> We thank the reviewer for pointing this out. The word “establish” was indeed an overstatement, without data on the correction of immature glycosylation and cholesterol accumulation. Following the reviewer’s advice, we toned down the statement to “…, supporting our hypothesis that itraconazole acts as a pharmacological chaperone for NPC1” as in the abstract.

---

## [Editor Report · Decision Letter 2]

26 Nov 2020

Image-based screen capturing misfolding status of Niemann-Pick type C1 identifies potential candidates for chaperone drugs

PONE-D-20-11428R2

Dear Dr. Ohgane,

We’re pleased to inform you that your manuscript has been judged scientifically suitable for publication and will be formally accepted for publication once it meets all outstanding technical requirements.

Kind regards,

Oscar Millet

Academic Editor

PLOS ONE
---

## [Editor Report · Acceptance letter]

4 Dec 2020

PONE-D-20-11428R2 

Image-based Screen Capturing Misfolding Status of Niemann-Pick Type C1 Identifies Potential Candidates for Chaperone Drugs 

Dear Dr. Ohgane:

I'm pleased to inform you that your manuscript has been deemed suitable for publication in PLOS ONE. Congratulations! Your manuscript is now with our production department. 

Kind regards, 

on behalf of

Dr. Oscar Millet 

Academic Editor

PLOS ONE